# Recurrent photic zone euxinia limited ocean oxygenation and animal evolution during the Ediacaran

Wang Zheng [1], Anwen Zhou[1,2], Swapan K. Sahoo [3] ✉, Morrison R. Nolan[4], Chadlin M. Ostrander [5,6], Ruoyu Sun [1], Ariel D. Anbar [7,8], Shuhai Xiao [4] & Jiubin Chen [1] ✉

The Ediacaran Period (~635–539 Ma) is marked by the emergence and diversification of complex metazoans linked to ocean redox changes, but the processes and mechanism of the redox evolution in the Ediacaran ocean are intensely debated. Here we use mercury isotope compositions from multiple black shale sections of the Doushantuo Formation in South China to reconstruct Ediacaran oceanic redox conditions. Mercury isotopes show compelling evidence for recurrent and spatially dynamic photic zone euxinia (PZE) on the continental margin of South China during time intervals coincident with previously identified ocean oxygenation events. We suggest that PZE was driven by increased availability of sulfate and nutrients from a transiently oxygenated ocean, but PZE may have also initiated negative feedbacks that inhibited oxygen production by promoting anoxygenic photosynthesis and limiting the habitable space for eukaryotes, hence abating the long-term rise of oxygen and restricting the Ediacaran expansion of macroscopic oxygen-demanding animals.

The Ediacaran Period (635–539 million years ago, or Ma) was a pivotal timeframe in the evolution of life on Earth, evidenced by the initial appearance of macroscopic multicellular eukaryotes in the fossil record (i.e., the "Ediacara biota") and its subsequent decline[1]. The rise and decline of the Ediacara biota are among the biggest mysteries in the evolution of life. Numerous studies have linked the rise of multicellular eukaryotes and the diversification of the Ediacara biota to ocean oxygenation[2,3]. Likewise, the decline of the Ediacara biota has been linked to a drop in ocean oxygen levels during the terminal Ediacaran Period[4,5]. However, the causal relationships between biological innovation and ocean redox during this period remain debated because of disparate ocean redox conditions reconstructed by different paleoredox proxies[6]. Redox-sensitive element (RSE) enrichments and carbon and sulfur isotope variations found in the

sedimentary record are interpreted as evidence of widespread oxic conditions, and perhaps even ventilation of the deep ocean, at least episodically during the Ediacaran[2,3,7,8]. Yet, Fe speciation data indicate persistently anoxic deep waters throughout most of the Ediacaran[9,10]. In recent years, a growing body of evidence supports a redox-stratified Ediacaran ocean, with some continental shelf margins overlain by a dynamic mid-depth euxinic water mass ("euxinic wedge")[11,12]. In this scenario, early animals could have thrived in oxic shelf settings even as deep oceans remained anoxic[13,14]. It was even suggested that surface ocean environments that met the oxygen requirements of the earliest metazoans (0.5–4.0% of present atmospheric levels) were already established well before the Ediacaran[15], leading to the proposal that the radiation of animals was not prohibited by oxygen availability. Thus, additional information about

[1]School of Earth System Science, Institute of Surface-Earth System Science, Tianjin University, Tianjin 300072, China. [2]Department of Earth, Ocean and Atmospheric Science and National High Magnetic Field Laboratory, Florida State University, Tallahassee, FL 32306, USA. [3]Equinor US, Houston, TX, USA. [4]Department of Geosciences, Virginia Tech, Blacksburg, VA 24061, USA. [5]Department of Marine Chemistry and Geochemistry, Woods Hole Oceanographic Institution, Woods Hole, MA 02543, USA. [6]Department of Geology and Geophysics, University of Utah, Salt Lake City, UT 84112, USA. [7]School of Earth and Space Exploration, Arizona State University, Tempe, AZ 85287, USA. [8]School of Molecular Sciences, Arizona State University, Tempe, AZ 85287, USA. ✉e-mail: swas@equinor.com; jbchen@tju.edu.cn

ocean redox conditions and their changes could help to resolve these complexities.

In this study, we refine the redox evolution in the Ediacaran ocean using mercury (Hg) stable isotopes. In particular, we focus on evidence for the presence of anoxic and $H_2S$-rich waters in the marine photic zone, a condition referred to as photic zone euxinia (PZE)[16,17]. Different from the previously proposed "euxinic wedge" that refers to a mid-depth euxinic water mass[11,12], PZE emphasizes the shallowing of the euxinic water mass into the photic zone, which is particularly detrimental to shallow marine inhabitants and has been proposed as a potent kill mechanism during almost all Phanerozoic mass extinction events[17] as well as a key factor responsible for the evolutionary stasis during the mid-Proterozoic[18]. The shelf areas are believed to have been major habitats for early animals[19], and thus the development of PZE would have shrunk the oxygenated habitats and possibly contributed to the delayed rise and eventual decline of the Ediacara biota. Moreover, PZE is typically found to be driven by enhanced nutrient influx to the ocean[20], which is exactly the mechanism proposed to trigger the oxygenation events in the Ediacaran ocean[7,8]. Therefore, there could be a potential link between ocean oxygenation and PZE, and thus the investigation on PZE may provide insights into the cause and consequence of ocean oxygenation in the Ediacaran Period.

Mercury (Hg) isotopes in sedimentary rocks are an emerging proxy for tracing paleoenvironmental changes such as large volcanic emission[21], as well as PZE[16]. A previous study has shown that marine sediments deposited under PZE display distinct negative excursions of mass independent isotope fractionation (MIF) of Hg relative to non-PZE conditions, demonstrating that Hg isotopes are a promising proxy for changes in ocean redox conditions[16]. Here we report Hg concentration and isotope ratios of the Ediacaran Doushantuo Formation in South China (Fig. 1). We studied four black-shale dominated sections, including Weng'an (WA, shallow shelf), Taoying (TY, upper slope), Wuhe (WH, lower slope), and Yuanjia (YJ, basin) sections, which represent a shelf-to-basin transect. The Doushantuo Formation (~635–560 Ma) is known for its exceptionally well-preserved fossil and geochemical record, and has been extensively studied to reconstruct the Ediacaran ocean environments[22]. Previous studies on the same

suite of black shale sections reported evidence for three ocean oxygenation events (OOEs) at ~635 Ma, ~580 Ma, ~565 Ma based on a variety of redox proxies (i.e., Fe speciation, RSE and $\delta^{34}S_{pyrite}$)[7,8]. Various metal isotopes ($\delta^{98}Mo$, $\delta^{53}Cr$, and $\varepsilon^{205}Tl$) have also been measured in these samples[23–25]. In the context of the published data, our Hg isotope data provide strong evidence for recurrent PZE as well as its impact on the oceanic redox evolution and biological innovation during this critical period.

## Results and discussion

### Hg concentration and isotope compositions

In WH, the total Hg concentrations (THg) are >100 ppb except for only two samples. Member IV, which correlates with the previously proposed OOE interval, shows notably higher THg than Member II and III (Fig. 2). To constrain the mechanism of Hg enrichment in sediments, THg is often normalized to major host phases of Hg, including organic matter (proxied by total organic carbon, TOC), sulfide (proxied by pyrite S, $S_{py}$) and clay minerals (proxied by Al concentration)[26]; the ratios of Hg/TOC, Hg/$S_{py}$ and Hg/Al are all significantly elevated in the Member IV OOE interval of WH (Fig. 2). The TY section, where only the three OOE intervals were analyzed, shows in general the highest THg among all sections and an increasing trend of THg and Hg/TOC from Member II to IV. YJ and WA sections were only analyzed for the Member II OOE interval. In YJ, THg and normalized ratios (Hg/TOC, Hg/$S_{py}$ and Hg/Al) show a transient elevation at the lowermost of the section, and decreases up-section. In WA, the variations of THg and normalized ratios show no apparent trend. The THg of different intervals is correlated with different host phases, and these correlations are described in details in Supplementary Text S1 and Supplementary Fig. S1.

The $\Delta^{199}Hg$ values (representing MIF of odd isotopes) of WH show cyclic variability, with distinct negative excursions at the lowermost Member II and III, as well as Member IV (down to −0.10‰ in the lowermost Member II), coinciding with three previously proposed OOE intervals[8], and gradual positive shifts (up to 0.19‰) between OOEs (Fig. 2). To facilitate discussion, we label the three intervals that show negative excursions of $\Delta^{199}Hg$ in Member II, III and IV of WH and the

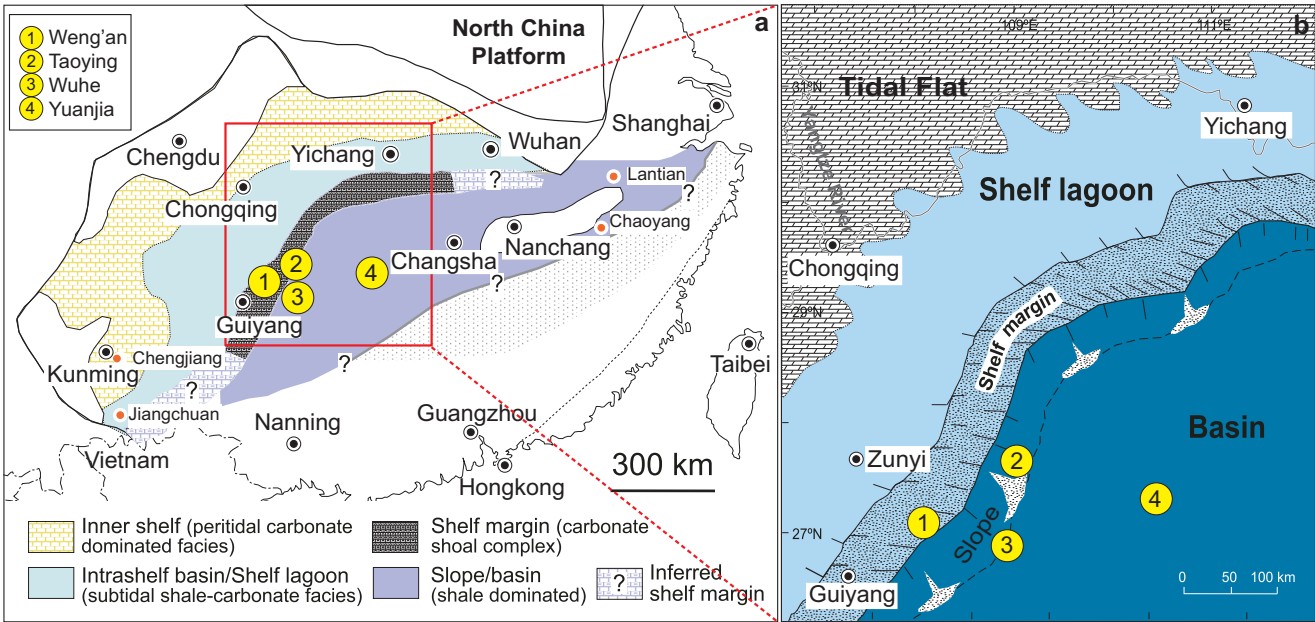

**Fig. 1 | Paleogeographic location of the study sections in the Yangtze Platform during the deposition of the Doushantuo Formation in South China.** The region marked by the red rectangle in panel **a** corresponds to panel **b**. The numbers in yellow circles are the study sections: Weng'an (WA), Taoying (TY), Wuhe (WH),

Yuanjia (YJ). Reprinted from Jiang, G., Shi, X., Zhang, S., Wang, Y. & Xiao, S. Stratigraphy and paleogeography of the Ediacaran Doushantuo Formation (ca. 635–551 Ma) in South China. *Gondwana Res.* 19, 831–849 (2011), with permission from Elsevier.

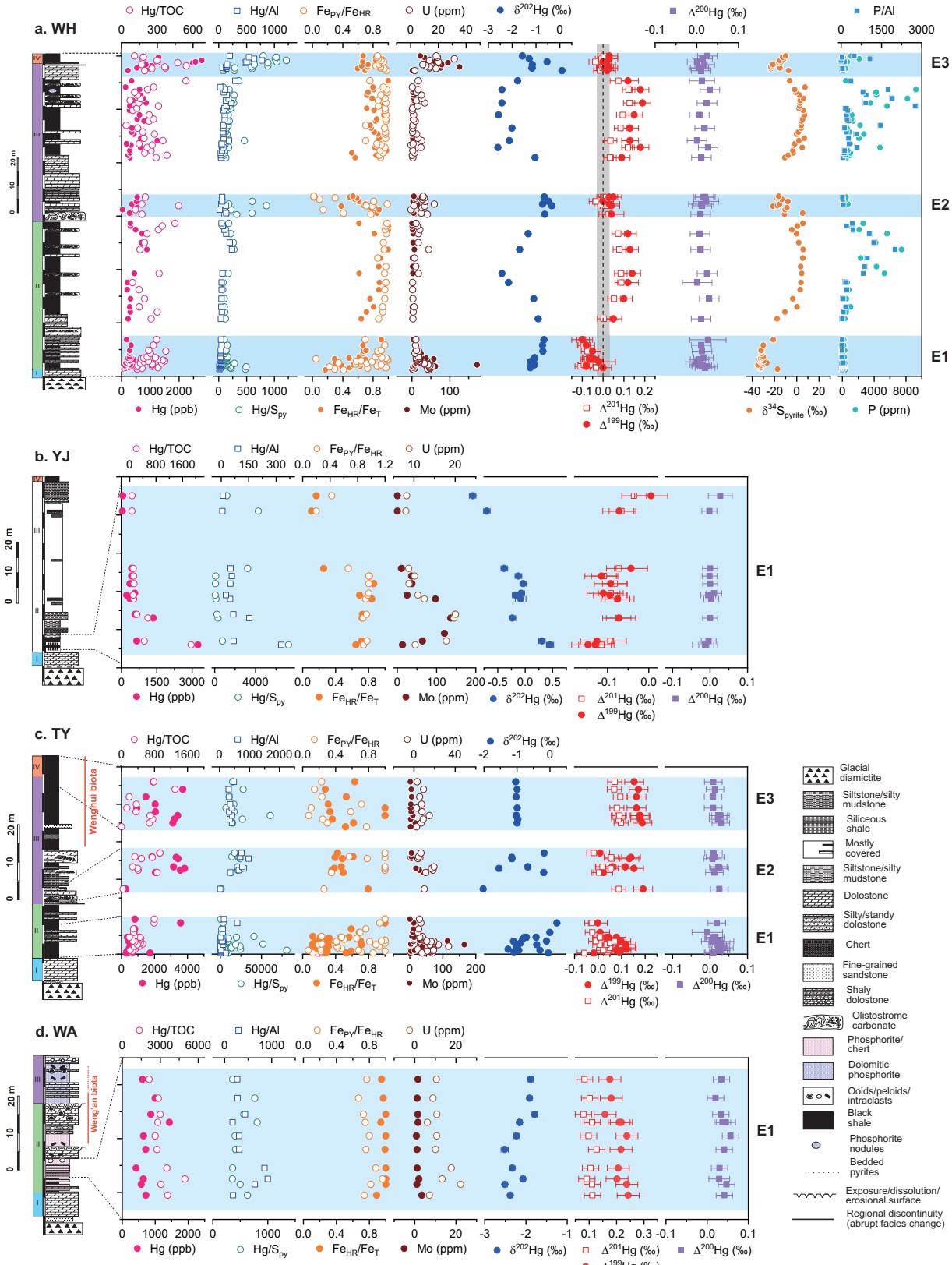

**Fig. 2 | Chemostratigraphy of all study sections of the Doushantuo Formation in South China. a** Wuhe (WH), **b** Yuanjia (YJ), **c** Taoying (TY) and **d** Weng'an (WA) sections. Fe speciation, redox sensitive elements (U, Mo), δ³⁴Sₚᵧᵣᵢₜₑ, P concentration and P/Al are from ref. 8. "TOC" stands for total organic carbon. The horizontal blue bands mark the intervals of previously proposed ocean oxygenation event (corresponding to the E intervals). The gray band in the Δ¹⁹⁹Hg and Δ²⁰¹Hg column of panel **a** represents the analytical uncertainty of Hg odd-mass independent fractionation (±0.04‰). The error bars for Hg isotopes are analytical uncertainties defined in the method section.

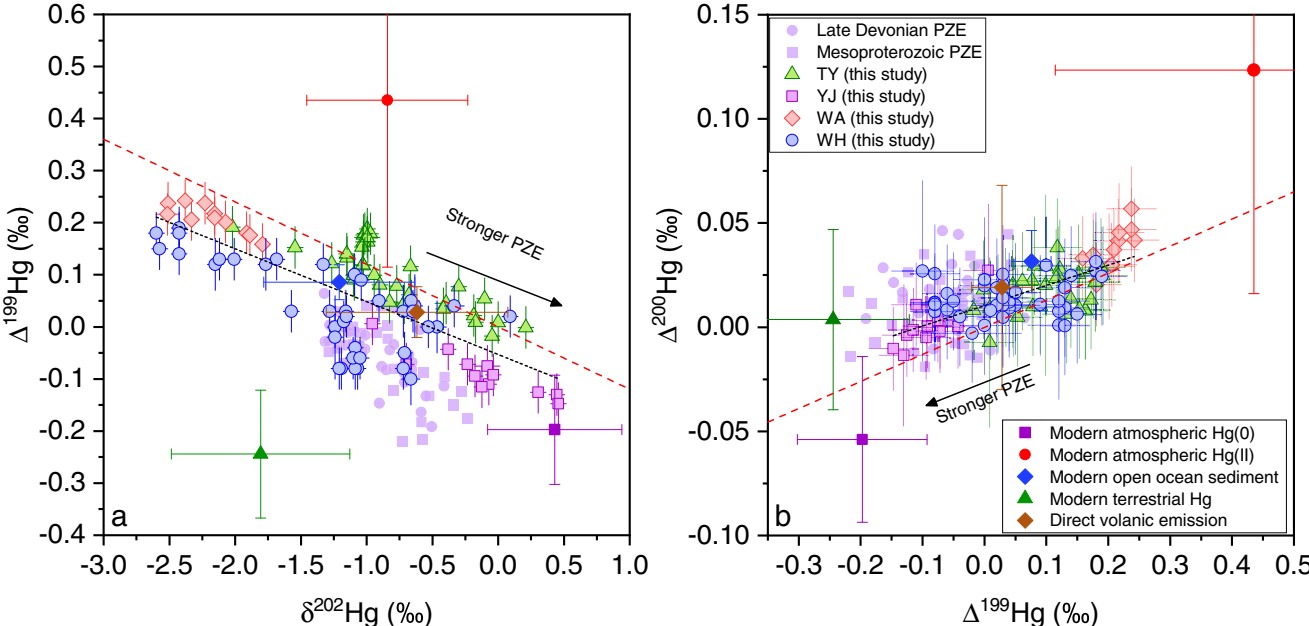

**Fig. 3 | Stable isotope compositions of Hg in all study sections. a** $\Delta^{199}$Hg vs. $\delta^{202}$Hg and **b** $\Delta^{200}$Hg vs. $\Delta^{199}$Hg for Weng'an (WA), Taoying (TY), Wuhe (WH), and Yuanjia (YJ) sections. Common Hg sources based on modern samples (average ±1 SD) are also plotted to show their possible contributions to Hg isotope variations of the Doushantuo samples. See Supplementary Fig. S5 and Supplementary Text S3 for details and references for modern data. The "Meso-proterozoic PZE" and "Late Devonian PZE" are the Hg isotope data in sedimentary rocks deposited under photic zone euxinia (PZE) from the Mesoproterozoic

Atar and El Mreiti groups[16], and from the Late Devonian Chattanooga Shale[39], respectively. The black short-dash lines are the linear regressions for all Doushantuo samples (in panel **a**, slope = −0.10 ± 0.01, $R^2$ = 0.62, $P$ < 0.001; in panel **b**, 0.10 ± 0.01, $R^2$ = 0.55, $P$ < 0.001). The red dash lines are reference lines of experimental results of dark abiotic oxidation of Hg(0) by thiol compounds (in panel **a**, slope = −0.12; in panel **b**, slope = 0.13)[46]. The error bars are analytical uncertainties defined in the method section.

corresponding intervals in other sections as "E1", "E2" and "E3", respectively ("E" standards for excursion). In WH, E1 shows the strongest negative excursion (with $\Delta^{199}$Hg decreasing by as much as −0.29‰ relative to the highest value between the E intervals), whereas both E2 and E3 show smaller but still notable negative excursions (with $\Delta^{199}$Hg dropping by −0.19‰). The other deep-water section, YJ, also shows negative $\Delta^{199}$Hg values (down to −0.15‰) at the bottom of E1, which then gradually increase to 0.01‰ up section. In contrast, the two shallower sections, WA and TY, show mostly positive $\Delta^{199}$Hg in the E intervals (up to 0.24‰ and 0.19‰, respectively), which is similar to the positive values in WH between the E intervals.

The $\delta^{202}$Hg values (representing mass dependent fractionation, MDF) of WH show a strong negative correlation with $\Delta^{199}$Hg (the $\Delta^{199}$Hg/$\delta^{202}$Hg slope = −0.09 ± 0.02, $R^2$ = 0.47, $P$ < 0.001), with positive excursions (up to 0.09‰) concurrent with negative shifts in $\Delta^{199}$Hg, and gradual negative shifts (down to −2.60‰) concurrent with positive shifts in $\Delta^{199}$Hg (Figs. 2 and 3). The WA, TY and YJ sections also show negative correlations between $\Delta^{199}$Hg and $\delta^{202}$Hg with similar $\Delta^{199}$Hg/$\delta^{202}$Hg slopes as in WH (Supplementary Fig. S2). Collectively, all four sections reveal a coherent linear $\Delta^{199}$Hg/$\delta^{202}$Hg correlation with a slope of −0.10 ± 0.01 ($R^2$ = 0.62, $P$ < 0.001) (Fig. 3a).

In addition to $\Delta^{199}$Hg/$\delta^{202}$Hg, the relationships between different MIF values ($\Delta^{200}$Hg/$\Delta^{199}$Hg and $\Delta^{199}$Hg/$\Delta^{201}$Hg) are also diagnostic of the mechanism of Hg isotope fractionation. The variation of $\Delta^{200}$Hg (representing MIF of even isotopes) in each single section is within the analytical uncertainty, but when all sections are combined, significant positive relationships between $\Delta^{200}$Hg and $\Delta^{199}$Hg, and hence negative correlations between $\Delta^{200}$Hg and $\delta^{202}$Hg can be observed, with a $\Delta^{200}$Hg/$\Delta^{199}$Hg slope of 0.10 ± 0.01 (1SE, $R^2$ = 0.55, $P$ < 0.001) (Fig. 3b) and a $\Delta^{200}$Hg/$\delta^{202}$Hg slope of −0.01 ± 0.00 (1SE, $R^2$ = 0.46, $P$ < 0.001) (Supplementary Fig. S3). $\Delta^{199}$Hg and $\Delta^{201}$Hg also show a significant positive relationship with a $\Delta^{199}$Hg/$\Delta^{201}$Hg slope of 1.49 ± 0.09 ($R^2$ = 0.96, $P$ < 0.001) for all sections (Supplementary Fig. S4).

## The influence of Hg sources on Hg isotope compositions in Doushantuo shales

The Hg isotope compositions in the study sections are unlikely modified by diagenetic or metamorphic alterations (see a detailed discussion in Supplementary Text S2), and thus their variations should represent changes in the primary Hg isotope signals of contemporaneous seawater, which could be affected by Hg sources or fractionation processes in ocean. It is important to consider all patterns of Hg isotope fractionation (including the direction and extent of the shifts in both MIF and MDF values as well as their relationships), because together they provide "multi-dimensional" constraints on the mechanism of Hg isotope fractionation and thus allow us to constrain the processes driving these fractionations. Since WH comprises a continuous deposition of black shales from Member II to IV, most discussion below focuses on WH.

The main Hg sources to the modern ocean include atmospheric deposition and terrestrial input[21], which have distinguishable Hg isotope signatures (Supplementary Fig. S5 and Supplementary Text S3). Enhanced input of terrestrial Hg was often invoked to interpret negative shifts in $\Delta^{199}$Hg of sedimentary rocks[21], based on the fact that modern terrestrial Hg associated with organic matter (OM) derived from soil and biomass typically exhibits negative $\Delta^{199}$Hg[27]. However, enhanced terrestrial Hg input is unlikely the main mechanism for the negative $\Delta^{199}$Hg shift in the E intervals. Precambrian terrestrial biomass was very scarce compared to today, except fungus-like microorganism found in cryptic karstic environments[28], and thus the modern-type OM-associated terrestrial Hg was unlikely a major source of Hg. Before the emergence of land plants, the dominant form of terrestrial Hg was likely geogenic Hg released by weathering and erosion of continental rocks, which is characterized by near-zero MIF[29] and thus could not explain the significantly negative $\Delta^{199}$Hg values in E1 of WH and YJ sections. Although the near-zero $\Delta^{199}$Hg in E2 and E3 of WH seem to be consistent with that of geogenic Hg, the correlations between THg and

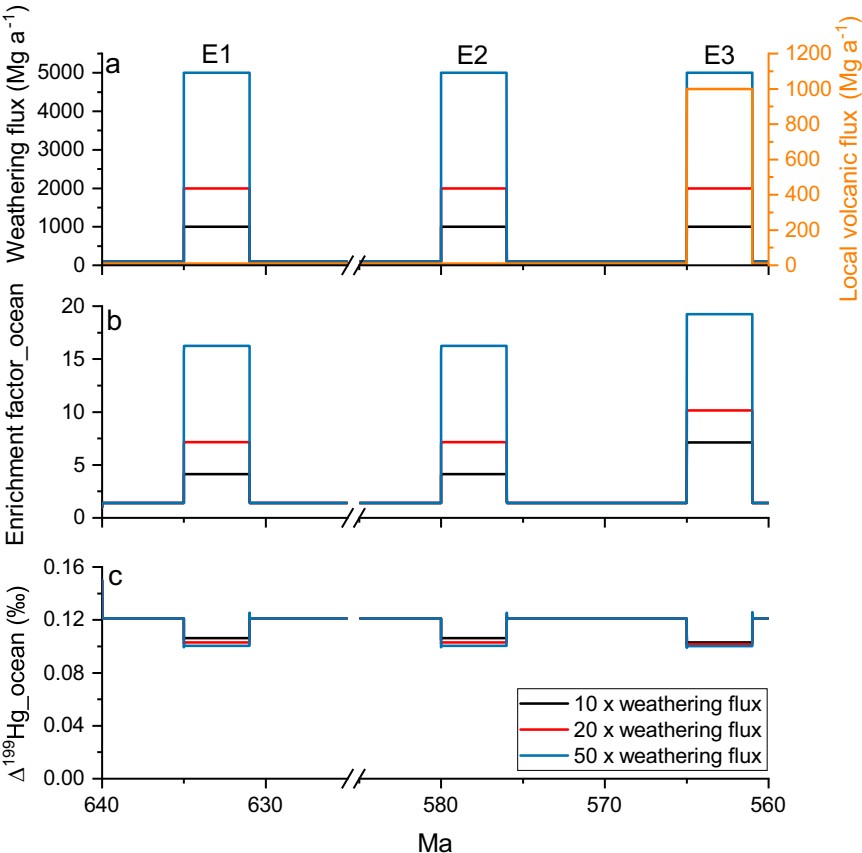

**Fig. 4 | Numerical simulation results using a simplified Hg isotope box-model.** Simulated marine Hg enrichment (**b**) and $\Delta^{199}$Hg (**c**) in response to an increase in terrestrial weathering Hg flux by 10–50× during E1 (-635 Ma), E2 (-580 Ma), and E3 (-565 Ma), and a further increase in local volcanic Hg emission by 100× during E3 (**a**). The "enrichment factor_ocean" value is defined as the marine Hg reservoir during E intervals relative to the reservoir size between E intervals.

Al (a proxy of continental weathering input) are insignificant in the entire WH section as well as in YJ and two E intervals of TY sections (Supplementary Text S1 and Supplementary Fig. S1), which strongly argue against enhanced terrestrial Hg input as the cause of the negative shift in $\Delta^{199}$Hg during E intervals. In addition, the Hg concentration in crustal igneous and metamorphic rocks is typically <10ppb[30,31], much lower than the >100ppb (and frequently >1000ppb) THg in almost all Doushantuo shales, which also argues against terrestrial Hg from continental weathering as a dominant source of Hg. Moreover, we simulated the shift in $\Delta^{199}$Hg caused by enhanced terrestrial weathering using a simplified Hg isotope box-model (see details of the model in Supplementary Text S4)[32], and found that even under a scenario of 50× increase in terrestrial Hg input, the $\Delta^{199}$Hg of the ocean can only be decreased by 0.02‰ (Fig. 4), which is an order of magnitude lower than the extent of $\Delta^{199}$Hg excursions during the E intervals (−0.19‰ to −0.29‰). The reason for the muted $\Delta^{199}$Hg shift under high terrestrial inputs is that the enhanced terrestrial input would increase the size of the oceanic Hg reservoir, which would also lead to enhanced re-emission of Hg from ocean to the atmosphere as observed in modern ocean[33,34]. The re-emitted Hg would undergo atmospheric redox transformations that produce a net positive $\Delta^{199}$Hg signal for atmospheric Hg species[35], which then deposit back to surface ocean. Thus, the enhanced re-emission and re-deposition of Hg with positive $\Delta^{199}$Hg following enhanced terrestrial weathering eventually counteracted the negative shift in $\Delta^{199}$Hg caused by terrestrial input. A full investigation on the effects of Hg re-emission and re-deposition on marine Hg isotope signatures is beyond the scope of the current study and is being undertaken in a separate study. Nevertheless, the modeling results further strengthen our argument that terrestrial Hg could not account for the negative $\Delta^{199}$Hg shift during E intervals.

Atmospheric Hg deposition is the dominant Hg source to global ocean in modern environment[36], and thus was likely a major Hg source in the Ediacaran ocean as well. The Hg isotope compositions of modern open ocean seawater are proposed to reflect mixing between atmospheric Hg(II) and Hg(0) deposition[36], which have opposite isotope signatures, with positive $\Delta^{199}$Hg, positive $\Delta^{200}$Hg and negative $\delta^{202}$Hg for the former, and negative $\Delta^{199}$Hg, negative $\Delta^{200}$Hg and positive $\delta^{202}$Hg for the latter (Supplementary Fig. S5 and reference therein). Therefore, a simple explanation for the cyclic variation of Hg isotopes in WH is a change in the proportion of atmospheric depositions, i.e., with a higher proportion of Hg(0) deposition during intervals with relatively more negative MIF (the E intervals) and a higher proportion of Hg(II) deposition during intervals with more positive MIF (between the E intervals). However, this simple interpretation is inconsistent with several observations. First, the $\Delta^{199}$Hg/$\Delta^{201}$Hg of both atmospheric Hg(II) and Hg(0) is -1.0 (typically between 1.0 and 1.1)[29], whereas the $\Delta^{199}$Hg/$\Delta^{201}$Hg of Doushantuo shales is -1.5 (Supplementary Fig. S4), which cannot be solely explained by the mixing of atmospheric Hg(II) and Hg(0). Second, changes in the proportions of atmospheric deposition to modern ocean have been shown to cause corresponding variation of $\Delta^{200}$Hg by more than 0.1‰ in marine sediments[36], because of the opposite $\Delta^{200}$Hg for atmospheric Hg(0) and Hg(II) (Supplementary Fig. S5), but this is inconsistent with the invariant $\Delta^{200}$Hg in WH (Fig. 2).

In addition to terrestrial input and atmospheric deposition, upwelling of Hg associated with OM (Hg-OM) from the deep ocean has also been proposed to drive the Hg isotope variation in the Doushantuo Formation by a previous study on coeval shallow shelf lagoon sections (e.g., Jiulongwan section)[37]. This study found primarily positive $\Delta^{199}$Hg shifts and negative $\delta^{202}$Hg shifts that are roughly in sync

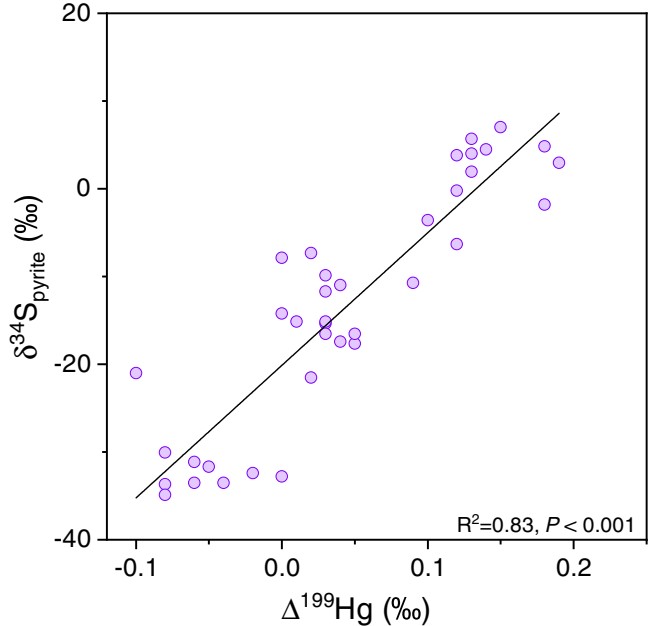

**Fig. 5 | Positive correlation between $\Delta^{199}$Hg and $\delta^{34}$S$_{pyrite}$ in Wuhe section.** The black solid line represents the linear regression between $\Delta^{199}$Hg and $\delta^{34}$S$_{pyrite}$ ($R^2 = 0.83$, $P < 0.001$).

with the negative carbonate carbon isotope ($\delta^{13}$C$_{carb}$) excursion in the cap dolostone (Member I) and Member IV in shelf lagoon sections. These Hg isotope signals were proposed to reflect the Hg released during the oxidation of deep ocean OM that is enriched in light carbon isotopes[37]. The isotopic signatures of Hg associated with deep ocean OM are hypothesized to resemble open ocean sediments[38]. Since the oxidation of OM is oxygen-demanding, this source should be more prominent during the proposed OOE intervals and in shallow water sections where the upwelling DOC meets the chemocline and is oxidized. In our study sections, Hg-OM is unlikely a dominant source for deep-water sections (WH and YJ), because they actually show opposite shifts in $\Delta^{199}$Hg and $\delta^{202}$Hg during the E intervals than the purported Hg-OM isotope signature (Fig. 2). E1 of the two shallower sections (WA and TY) shows primarily positive $\Delta^{199}$Hg that is similar to modern open ocean sediments, and strong correlations between THg and TOC (Supplementary Text S1 and Supplementary Fig. S1), indicating possible contributions of Hg-OM. However, Hg-OM alone still cannot explain the high $\Delta^{199}$Hg/$\Delta^{201}$Hg slope for all sections (-1.5) (Supplementary Fig. S4), which is significantly higher than those of open ocean sediments (close to 1.0)[38].

Therefore, while terrestrial Hg, atmospheric Hg, and deep ocean Hg-OM may all constitute the sources of Hg in the Doushantuo shales, the variations of Hg isotopes could not be fully explained by changes in or mixing of these Hg sources, but require fractionation of Hg isotopes in ocean.

## Hg isotope evidence for PZE in the Ediacaran ocean

The distinct negative excursions of $\Delta^{199}$Hg with concurrent positive excursions of $\delta^{202}$Hg during the E intervals were most likely driven by Hg isotope fractionation in seawater under PZE conditions, and this hypothesis is supported by abundant and compelling evidence. First, there exists a strong positive correlation between $\Delta^{199}$Hg and pyrite S isotopes ($\delta^{34}$S$_{pyrite}$) in WH section ($R^2 = 0.83$, $P < 0.001$) (Fig. 5). Very negative $\delta^{34}$S$_{pyrite}$ values (as low as −34.9‰) are found during the proposed OOE intervals at WH and have been attributed to an increased availability of marine sulfate, which permits the expression of large-magnitudes of S isotope fractionation during bacterial sulfate reduction[7]. Increased marine sulfate availability would also increase

H$_2$S production, fueling the expansion of euxinia. Thus, the highly synchronized shifts in $\Delta^{199}$Hg and $\delta^{34}$S$_{pyrite}$ suggest a first-order control of redox conditions on the variation of Hg isotopes. Furthermore, the THg of WH shows the strongest correlation with S$_{py}$ among all host phases in at least two E intervals (corresponding to Member II and Member IV OOEs) (Supplementary Text S1 and Supplementary Fig. S1), suggesting that Hg was primarily scavenged to sediments by sulfide minerals. The strong correlation between THg and sulfide is typically interpreted as a sign of euxinic depositional environment[26]. In contrast, between the E intervals, THg shows a stronger correlation with TOC than with S$_{py}$ (Supplementary Text S1 and Supplementary Fig. S1), suggesting that the sedimentation of Hg was controlled by OM, like in modern open oceans[21]. Thus, the change in host phases from sulfide to OM likely indicates a transition from locally euxinic to non- or less euxinic conditions.

Second, the patterns of Hg isotope excursions (including the direction and extent of the shifts in both $\Delta^{199}$Hg and $\delta^{202}$Hg values as well as $\Delta^{199}$Hg/$\delta^{202}$Hg) in the E intervals are strikingly similar to the Hg isotope systematics from two previously reported successions where PZE was validated independently. Zheng et al. reported Hg isotopic data from Mesoproterozoic black shales (-1.1 Ga) of the Atar and El Mreiti groups, West Africa[16], and from the Late Devonian Chattanooga Shale, North America[39], and both display comparable negative shifts in $\Delta^{199}$Hg (by as much as ~ −0.2‰) during PZE with similar negative $\Delta^{199}$Hg/$\delta^{202}$Hg of −0.15 and −0.10, respectively (Fig. 3). The PZE condition in the two above-mentioned studies was independently validated by organic biomarkers produced by anoxygenic phototrophic bacteria that metabolize H$_2$S[40,41]. While indigenous biomarkers are not yet available for the Doushantuo shales[42], the highly similar Hg isotope signatures between these successions serves as compelling evidence of a shared driving mechanism.

As proposed previously, PZE can result in concurrent negative $\Delta^{199}$Hg shift and positive $\delta^{202}$Hg shift via two mechanisms[16]. The first is photoreduction of Hg(II) bound to reduced organic sulfur ligands [i.e., Hg(II)-S] in a sulfide-rich photic zone. Under sulfidic conditions, dissolved organic matter in seawater will likely have a high fraction of thiols due to sulfurization[43], favoring the formation of Hg(II)-S complexes. Multiple experimental studies have demonstrated that the photoreduction of Hg(II)-S increases $\delta^{202}$Hg and decreases $\Delta^{199}$Hg in the remaining Hg(II) phase[44,45], which is consistent with the pattern of Hg isotope shifts observed in the E intervals. One caveat is that the experimental $\Delta^{199}$Hg/$\delta^{202}$Hg slope of this photoreduction process (-−0.7) is different from that of the Doushantuo shales (−0.10)[44], suggesting that the photoreduction of Hg(II)-S is not the only process involved.

The second mechanism is enhanced oxidation of atmospheric Hg(0) in seawater induced by the abundant thiol compounds under PZE conditions. This mechanism is fully compatible with the Hg isotope signatures of Doushantuo shales. Our previous experimental study shows that thiol-induced aqueous Hg(0) oxidation leads to negative shifts in $\Delta^{199}$Hg (by as much as −0.20‰), positive shifts in $\delta^{202}$Hg and no changes in $\Delta^{200}$Hg in the oxidized Hg(II) phase, with a characteristic $\Delta^{199}$Hg/$\delta^{202}$Hg of −0.12 and a $\Delta^{199}$Hg/$\Delta^{201}$Hg of 1.6[46], which are all fully consistent with the pattern of Hg isotope variation in the E intervals (Fig. 3 and Supplementary Fig. S2). Although it may seem counter-intuitive that Hg(0) oxidation is enhanced in sulfidic water that is typically considered as a reducing environment, multiple experimental studies indeed show that aqueous Hg(0) oxidation can be enhanced by either free thiol or thiol ligands in organic matter[46,47]. Sulfide also inhibits the re-emission of Hg(0) from seawaters by complexing and scavenging the oxidized Hg to sediments. Thus, the presence of PZE could promote the marine uptake of atmospheric Hg(0) by enhancing its oxidation in the surface ocean and subsequent sequestration to sediments. Moreover, the rate of Hg(0) oxidation in sulfidic waters determined by experiments is comparable to many

other major redox pathways of Hg in natural aquatic systems (e.g., photooxidation and photoreduction)[47], making the thiol-induced Hg(0) oxidation an environmentally relevant process and potentially the key mechanism that caused the observed shifts in Hg isotopes during periods of PZE.

In contrast to the E intervals, the gradual positive shifts in $\Delta^{199}$Hg with concurrent negative shifts in $\delta^{202}$Hg between the E intervals indicate the contraction of PZE and the transition to modern-like oxygenated surface ocean, which is supported by the change in Hg host phases from sulfide to OM (Supplementary Fig. S1), as well as the similarly positive $\Delta^{199}$Hg and negative $\delta^{202}$Hg in modern ocean sediments[36,38,48] (Supplementary Text S3 and Supplementary Fig. S5). Note that this interpretation does not conflict with the Fe speciation data between the E intervals ($Fe_{HR}/Fe_T > 0.38$, $Fe_{py}/Fe_{HR} > 0.8$) (Fig. 2), which is typically interpreted to reflect local bottom water euxinia[49]. Sahoo et al. who first reported these Fe speciation data suggested that the high $Fe_{py}/Fe_{HR}$ could potentially reflect sulfidic conditions in sediment porewater rather than the water column[7]. In contrast, Hg isotopes specifically track euxinia in the photic zone, because the two proposed mechanisms controlling Hg isotope fractionation under PZE both operate in surface water where light can penetrate or atmospheric gaseous Hg(0) can dissolve. There is currently no known mechanism that produces characteristic Hg isotope signatures in euxinic (but non-PZE) bottom water or sediment porewater, where sulfide only acts as a ligand that binds with Hg. In fact, the study by Zheng et al.[16] found that the Mesoproterozoic deeper-water section deposited under euxinic (but not necessarily PZE) conditions shows no or a much weaker negative shift in $\Delta^{199}$Hg than the shallower section where PZE occurred, suggesting that Hg isotopes track specifically PZE rather than euxinia in general.

### Spatial variation of PZE in the South China Basin

The Hg isotope chemostratigraphic data are incomplete for three of the four study sections (YJ, TY, and WA) because they are not completely preserved, thus we are unable to perform a "full profile" reconstruction for redox conditions in these sections as for WH. However, all four sections have Hg isotope data for E1 and they show clear variations of Hg isotopes within E1. By comparing the variation patterns of Hg isotopes during E1 across the four sections, which cover a transect from the shelf to the basin, we are able to evaluate the spatial extent of PZE and the dynamics of the euxinic water mass within the Ediacaran South China Basin. The E1 of WH also shows the strongest negative shift in $\Delta^{199}$Hg among all E intervals, possibly indicating the most extensive PZE event. Thus, our discussion below focuses on E1.

Among the four study sections, the basinal section (YJ) shows the most negative $\Delta^{199}$Hg and most positive $\delta^{202}$Hg at the lowermost E1, and it also exhibits an up-section positive shift in $\Delta^{199}$Hg and concurrent negative shift in $\delta^{202}$Hg with a $\Delta^{199}$Hg/$\delta^{202}$Hg ratio similar to that in WH, suggesting that PZE was likely the most extensive at the onset of E1 but weakened towards the end of E1 at YJ (Fig. 2). In contrast, both the lower-slope (WH) and upper-slope (TY) sections show clear up-section negative shifts in $\Delta^{199}$Hg and concurrent positive shifts in $\delta^{202}$Hg with similar $\Delta^{199}$Hg/$\delta^{202}$Hg ratios, suggesting a gradual expansion of PZE at these two sites. The WA section located on the shallow shelf margin has the most positive $\Delta^{199}$Hg and the most negative $\delta^{202}$Hg among the four sections, and its THg correlates with TOC and Al, but not with sulfide (Supplementary Text S1 and Supplementary Fig. S1), suggesting that WA was likely located above the euxinic water mass or the PZE at WA was too sporadic to be captured by the current data. This is in agreement with previous paleogeographic reconstruction of the Doushantuo Formation in South China, which suggests that WA was likely exposed to oxic/suboxic water above the chemocline[22].

Based on the above observations, we can reasonably propose that PZE was widespread during the deposition of the basal Doushantuo

shale in South China, extending at least from the upper slope to the basin, which is consistent with the previous proposal for a mid-depth euxinic wedge on continental shelves of the South China Basin[11,12]. Furthermore, the euxinic wedge probably became shallower approaching the end of E1, and this is evident from the different patterns of Hg isotope variations within E1 across different sections. As discussed above, PZE was weakened at the deepest YJ section towards the end of E1, while PZE intensified at the shallower WH and TY sections, suggesting an up-slope migration of the euxinic water mass toward the end of E1 as depicted by Fig. 6. It is worth noting that E1 coincided with a sea level highstand[23]. Thus, the up-slope migration of the euxinic wedge towards the end of E1 could be associated with a global or local sea level change, although this hypothesis needs to be tested further.

### Plausible drivers of PZE

The occurrence of PZE during the previously proposed OOE intervals was likely a result of increased marine sulfate and nutrient availability, which has been previously proposed to trigger the Member II OOE (~635 Ma, E1) due to enhanced continental weathering in the aftermath of the Marinoan glaciation[7]. During the process of deglaciation, it is hypothesized that a surge of nutrients from freshly exposed land surfaces and finely ground glacial tills was delivered to the ocean, which increased primary productivity and thus oxygen consumption by newly produced organic matter, leading to anoxia on a timescale of ~$10^4$ years[2,50]. The enhanced primary productivity would also increase the organic matter available for microbial sulfate reduction and hence seawater $H_2S$ accumulation in near-shore environments, leading to PZE. On a longer timescale (~$10^6$ years), increased nutrient influx would increase organic carbon burial and hence atmospheric oxygen[50]. A similar sequence-of-events may have followed the Gaskiers glaciation, which occurred broadly coeval with the Member III OOE (~580 Ma, E2)[8]. However, this link is speculative given the uncertain timing of the Member III OOE, and it merits further examination.

The increased availability of nutrients in global oceans after the Marinoan glaciation is well-documented by the apparent expansion of the marine phosphorous (P) reservoir[51–54]. Phosphorous is a key limiting nutrient and increased P influxes to the ocean have been proposed to account for the development of PZE in many cases across Earth history, particularly during various Phanerozoic mass extinction events[55,56], and in modern meromictic lakes (e.g., the Mahoney Lake)[57]. Both P and P/Al (to account for changes in sedimentation rate) in the WH section show a cyclic variation similar to Hg isotopes and other redox proxies, with lower P during the E intervals and higher P in between (Fig. 2). The accumulation of P in sediments is controlled by local redox conditions, and euxinic conditions would re-dissolve and recycle P in sediments back to the water column due to rapid reduction of P-bearing Fe oxides by $H_2S$ and preferential release of P from remineralization of organic matter by sulfate-reducing bacteria[58]. Thus, the variation of sedimentary P is consistent with the local redox conditions at WH, and suggests intensive recycling and regeneration of P modulated by a rising marine sulfate reservoir[54], which may have stimulated PZE via positive feedback.

The Member IV OOE (E3) of both WH and TY sections shows higher Hg enrichments and different patterns of Hg isotopes compared to other OOEs, which we ascribe to Hg input from local volcanic emission (for WH) or upwelling of deep ocean Hg-OM (for TY) (see detailed discussion in Supplementary Text S5). Despite the influences of volcanic or upwelling Hg, the E3 interval of WH still shows a strong correlation between THg and $S_{py}$ (Supplementary Fig. S1). Furthermore, we also simulated the shift in $\Delta^{199}$Hg caused by enhanced local volcanic Hg input using the Hg isotope box-model (more details in Supplementary Text S4), and found that an increase in local volcanic Hg flux by 100× could only decrease $\Delta^{199}$Hg by <0.03‰ (Fig. 4), which is similar to the muted $\Delta^{199}$Hg shift caused by terrestrial Hg input. As

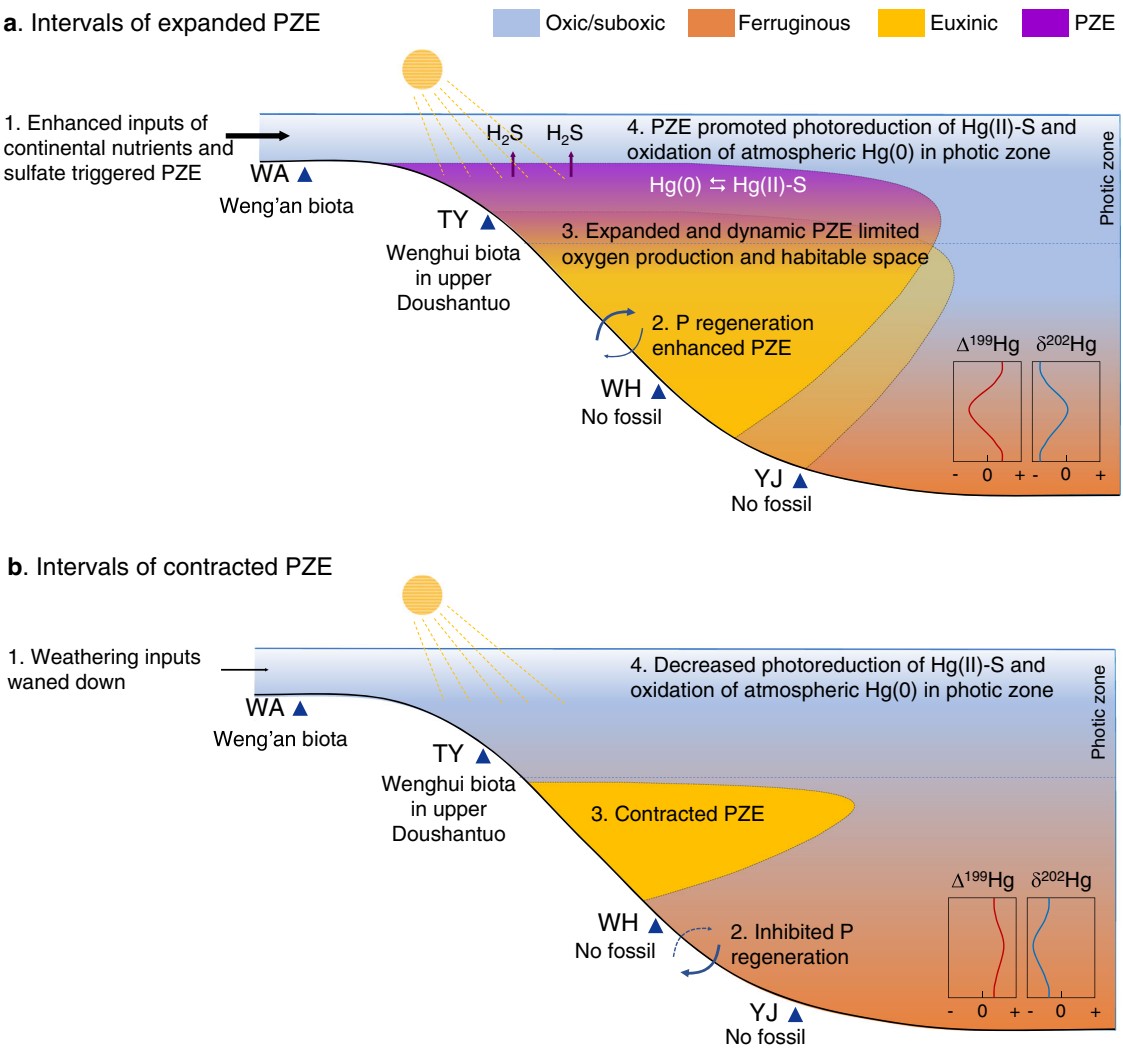

**Fig. 6 | Photic zone euxinia (PZE) in the Ediacaran ocean and its impact on ocean oxygenation and the Ediacara biota.** Conceptual shifts in $\Delta^{199}$Hg and $\delta^{202}$Hg under the depicted redox conditions are shown in inserted boxes. The numbers in text boxes represent the sequence of events. During intervals of expanded PZE (panel **a**), PZE developed on continental margins due to increased marine nutrients and sulfate availability (likely due to enhanced postglacial continental weathering, particularly during E1), and was further promoted by the positive feedback from P regeneration under euxinic water. PZE would have enhanced the photoreduction of Hg(II)-S ("S" represents reduced sulfur ligands) and the oxidation of atmospherically deposited Hg(0), leading to the negative shift in $\Delta^{199}$Hg with concurrent positive shift in $\delta^{202}$Hg. Expanded PZE also promoted anoxygenic photosynthesis and inhibited oxygenic photosynthesis, resulting in relatively rapid termination of the previously proposed ocean oxygenation event. Spatial variations in Hg isotopes indicate that PZE may have migrated upslope toward the end of E1, as depicted by the upper euxinic wedge in panel **a**. During intervals of contracted PZE (panel **b**), the postglacial surge of weathering may have waned down, leading to the contraction of PZE and resulting in the negative shift in $\delta^{202}$Hg with concurrent positive shift in $\Delta^{199}$Hg. PZE may have also limited the diversification of complex eukaryotic life. At Weng'an (WA) section that was not affected by PZE, the Weng'an biota was found in the upper phosphorite. At Taoying (TY) section where PZE was likely sporadic, the Wenghui biota was found in the upper Doushantuo black shales. In contrast, at Wuhe (WH) and Yuanjia (YJ) sections where PZE or anoxic water persisted, no benthic fossils have been reported.

explained earlier, the reason for the muted $\Delta^{199}$Hg shift might be due to the re-emission of volcanically-sourced Hg from ocean and the re-deposition of atmospheric Hg with positive $\Delta^{199}$Hg back to ocean. Thus, the negative excursion E3 with a drop in $\Delta^{199}$Hg by −0.19‰ may also suggest PZE. This hypothesis is consistent with the previous proposal for widespread euxinia on continental shelves based on V isotopes in black shales of Member IV[59]. Volcanism and upwelling may also lead to enhanced nutrient availability, which may trigger PZE, although this hypothesis requires further verification.

**The negative impacts of PZE on ocean oxygenation and the Ediacara biota**
PZE may have exerted a negative impact on the oxygenation of the Ediacaran ocean by promoting anoxygenic photosynthesis and inhibiting oxygenic photosynthesis (Fig. 6). This may be an

underappreciated mechanism for the transient nature of the Ediacaran OOEs. While oxygenic photosynthesis is the overwhelming source of free O2, it can be outcompeted by anoxygenic photosynthesis under conditions that favor the latter, such as extensive PZE[60,61] or ferruginous conditions[62]. Compelling experimental evidence has shown that even μM levels of H2S can completely shut down oxygenic photosynthesis[63]. It has been suggested that the competition for light and nutrients between anoxygenic and oxygenic photosynthetic organisms was a key mechanism responsible for the delayed oxygenation of Earth's atmosphere[62], and for the relatively low level of oxygen throughout the mid-Proterozoic[18]. Although our Hg isotope data only cover South China, lipid biomarkers for PZE have been reported in late-Ediacaran petroleum from Eastern Siberia[64], suggesting that PZE may have been common in various continental margins during the Ediacaran Period. Continental margins are the major

habitats for early primary producers such as cyanobacteria and ancestral photosynthetic eukaryotes[61]. Thus, we propose that the widespread PZE in continental margins initiated a negative feedback on oxygenic photosynthesis, which may be partially responsible for the relatively rapid termination of OOE. This argument is consistent with a recent hypothesis based on theoretical models that found the cyclic and oscillating oxygenation and deoxygenation events during the Ediacaran Period can be explained by internal feedbacks in the biogeochemical cycles of carbon, oxygen and phosphorous, without the need for specific external forcing[65]. As an important step forward, our finding of PZE provides an independent validation and mechanistic insights into this theoretical hypothesis.

Recurrent and widespread PZE in the Ediacaran ocean may have also inhibited the diversification and ecological expansion of early macroscopic eukaryotic life by limiting their habitable space via $H_2S$ poisoning. $H_2S$ toxification during PZE was a potent kill mechanism for some of the largest mass extinctions, such as the end-Permian[20,32,66] and end-Triassic mass extinctions[67]. Emerging geochemical data suggest a redox control on the rise and fall of the Ediacara biota[4,5,68]. The unstable redox conditions and repeated development of PZE in early Ediacaran oceans as revealed by Hg isotopes likely limited the ecological expansion of macroscopic multicellular eukaryotes. For example, the lower slope and basinal facies, as represented by the WH and YJ sections, were strongly influenced by PZE during the OOEs (the E intervals) and were consistently anoxic between OOEs, excluding the establishment of benthic multicellular eukaryotes that require free oxygen. In relatively shallow-water shelf and upper slope facies, as represented by the WA and TY sections, eukaryotes were able to colonize the benthic realm where and when PZE was absent. For example, multicellular eukaryote fossils of the Weng'an biota at WA occur in phosphorite strata correlated with Member II above the E1 interval[69]. Similarly, macroalgal fossils of the Wenghui biota at TY occur in black shales correlated to the Member III above the E2 interval (Fig. 2). These fossiliferous intervals represent locally oxic environments (as indicated by negative Ce anomalies in fossiliferous phosphorite[70]) that were habitable for early Ediacaran eukaryotes. A limitation of this study is that we did not directly measure samples from fossiliferous intervals. Thus, future studies that compare fossiliferous and non-fossiliferous intervals are needed to further verify the proposed impact of PZE on the Ediacara biota.

In summary, our study reveals recurrent, widespread and spatially dynamic PZE on the continental margin of South China during previously proposed OOE intervals of the Ediacaran Period. Although the presence of a dynamic euxinic wedge at South China during the Ediacaran Period has been proposed previously, our study shows strong evidence for the invasion of euxinic water mass into the marine photic zone, which has important implications for understanding its impacts on the redox evolution and biological innovation in the Ediacaran ocean. We suggest that widespread PZE at continental margins was a consequence of increased marine nutrient and sulfate reservoirs due to their positive feedback on primary productivity, but PZE may have also initiated a series of negative feedbacks that inhibited oxygen production (and thus decelerated ocean oxygenation) by promoting anoxygenic photosynthesis and limiting the habitable space for eukaryotes carrying out oxygenic photosynthesis. Therefore, PZE could be one of the key reasons for the transient nature of ocean oxygenation during the Ediacaran Period and the delayed rise of the Ediacara biota.

## Methods
### Study sections
The locations, stratigraphy, and paleogeography of the study sections are described in detail in previous work[7,8,22], and thus only a brief overview is presented here. The Doushantuo Formation is best known in the Yangtze Gorges area where it is divided into four lithostratigraphic members. Member I serves as a regional stratigraphic marker bed and corresponds to the cap carbonate overlying the terminal Cryogenian-age glacial diamictite of the Nantuo Formation. A zircon U-Pb date from a volcaniclastic layer atop Member I constrains its deposition age to be $635 \pm 0.6$ Ma[71]. Member II consists primarily of black shales with subordinate carbonate beds. Member III is dominated by carbonate although black shale becomes increasingly abundant toward deep-water facies outside the Yangtze Gorges area. The Member II-III boundary is correlated with a regional stratigraphic discontinuity in shallow-water facies[22], and if this regional discontinuity tracks a widespread marine regression, it may be time-equivalent with the Gaskiers glaciation (~580 Ma)[8]. Member IV is dominated by organic-rich black shales, and can also be used as a regional stratigraphic marker bed for the Doushantuo Formation in South China. The age of Member IV is a topic of ongoing debate, but it is probably ~565 Ma[72]. The shelf and upper slope sections (WA and TY) contain Ediacaran microfossils of the Weng'an biota[14] and macrofossils of the Wenghui biota[73], respectively. No fossils have been reported from the WH and YJ sections that represent lower slope and basinal facies.

### Hg concentration analysis
Mercury concentrations in solid samples were analyzed on a Lumex R-915F Hg Analyzer at Tianjin University. Mercury in samples was released as Hg(0) vapor by combustion at a temperature of ~750 °C, and then measured by a cold vapor atomic absorption spectroscopy (CV-AAS) with a detection limit lower than 0.5 ng/g. A certified reference material GBW07311 (GSD-11, freshwater sediment) was measured repeatedly alongside samples to monitor analytical accuracy and reproducibility. The GSD-11 yielded an average Hg concentration of $72.2 \pm 5.8$ ng/g (2 SD, $n = 8$), which is consistent with its certified value $72 \pm 9$ ng/g.

### Hg isotope analysis
Mercury isotopes were analyzed using multi-collector inductively coupled plasma mass spectrometry (MC-ICPMS, Neptune Plus, Thermo Fisher Scientific) at School of Earth System Science, Tianjin University based on published methods[74]. Prior to isotopic analysis, Hg in samples was first extracted by acid digestion and then purified by ion-exchange chromatography (see details in Supplementary Text S6)[16]. To monitor Hg yields and the accuracy of Hg isotope measurement, procedural blanks and three standard reference materials (SRM), including GBW07405 (Yellow/red soil), NIST 2702 (Inorganics in marine sediment), USGS SBC-1 (Brush Creek Shale), and the NIST SRM 3133 Hg isotope standard, were processed alongside samples. Recoveries for all SRM were $100 \pm 19\%$ (2 SD, $n = 19$) (Supplementary Table S1). Mercury concentrations in procedural blanks were typically less than 1% of the Hg present in samples.

Solutions eluted from the chromatographic procedure were diluted to 1.5 ng/g of Hg using a matrix solution containing 5% HCl. Thereafter, Hg in the diluted solutions was reduced by $SnCl_2$ (3%, w/v) to gaseous Hg(0), which was then carried into the plasma of MC-ICPMS by Hg-free argon gas. Simultaneously, thallium (Tl) aerosol (NIST SRM 997) generated by Aridus II desolvator was introduced together with Hg(0) vapor into the plasma. Five Hg isotopes ($^{198}Hg$, $^{199}Hg$, $^{200}Hg$, $^{201}Hg$, and $^{202}Hg$) and two Tl isotopes ($^{203}Tl$, $^{205}Tl$) were simultaneously measured via Faraday cups. Instrumental mass bias was corrected for using a combination of internal calibration with measured $^{205}Tl/^{203}Tl$ ratios and standard-sample-standard bracketing (relative to the NIST SRM 3133 Hg standard). The bracketing standard was matched to samples in terms of both matrix and Hg concentration (less than 10% difference). On-peak zero corrections were applied to all measured Hg masses. Mercury isotope compositions are reported using δ notation

defined by the following equation:

$$\delta^x Hg_{(‰)} = \left[ \frac{(^x Hg/^{198}Hg)_{sample}}{(^x Hg/^{198}Hg)_{std}} - 1 \right] \times 1000 \qquad (1)$$

where $^x Hg$ is $^{199}Hg$, $^{200}Hg$, $^{201}Hg$, or $^{202}Hg$, and "std" represents the NIST SRM 3133 standard. The MDF is reported as $\delta^{202}Hg$, and MIF is reported as the capital delta notation ($\Delta$) according to the following equation:

$$\Delta^x Hg_{(‰)} = \delta^x Hg - (\delta^{202}Hg \times \beta) \qquad (2)$$

where x is the mass number of Hg isotope 199, 200, and 201. $\beta$ is a scaling constant used to calculate the theoretical kinetic MDF, and it is 0.2520, 0.5024, and 0.7520 for $^{199}Hg$, $^{200}Hg$, and $^{201}Hg$, respectively[75].

To ensure data quality, each sample was measured at least twice, and a commonly used reference standard NIST 8610 was measured every 6 – 7 samples to monitor instrument performance. The averages of all NIST 8610 are: $\delta^{202}Hg = -0.54 \pm 0.06‰$, $\Delta^{199}Hg = -0.03 \pm 0.04‰$, $\Delta^{200}Hg = 0.00 \pm 0.02‰$ (2 SD, $n = 39$), consistent with the published values[75]. The NIST SRM 2702 yielded average $\delta^{202}Hg$, $\Delta^{199}Hg$ and $\Delta^{200}Hg$ values of $-0.77 \pm 0.04‰$, $-0.03 \pm 0.06‰$, and $0.00 \pm 0.02‰$ (2SE, $n = 3$), respectively (Supplementary Table S1). All isotope data are reported in Supplementary Table S2 and analytical uncertainties are reported as either 2 standard error (2SE) of sample replicates or 2 SD of all measurements of the NIST 8610, whichever is higher.

## Data availability

The authors declare that all data reported this study are available in the Supplementary Information file and at Figshare https://doi.org/10.6084/m9.figshare.23003246.

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

## Acknowledgements

This work was funded by the National Natural Science Foundation of China (grant No. 41973009, 41830647), National Key R&D Program of China (2022YFF0800300) and the National Science Foundation (EAR 2021207 awarded to S.X., and EAR 1760203 to A.D.A.).

## Author contributions

W.Z., S.K.S. and J.C. conceived, designed and supervised the study. A.Z. performed the analyses of Hg concentration and Hg isotopes. R.S. performed the modeling work. W.Z., J.C., S.X. and A.D.A. acquired funding for this study. W.Z. led the paper writing with significant inputs

from M.R.N., C.M.O., A.D.A., and S.X. All authors contributed to data interpretation and writing.

## Competing interests

The authors declare no competing interests.
