## [Peer Review File · Nature Communications]

REVIEWER COMMENTS

Reviewer #1 (Remarks to the Author):

Zheng and co-authors reported Hg concentration and isotope ratios in four black shale sections of the Ediacaran Doushantuo Formation in South China. And they confirmed widespread and spatially dynamic PZE on the continental margin of South China during at least two of the previously purported OOE intervals of the Ediacaran Period. The origins of PZE, both positive feedback on primary productivity and negative feedbacks that inhibited oxygen production, were discussed from regional and global scales. The study firstly provides more interesting and high-quality Hg isotope data during early-middle Ediacaran period, but also add some new knowledge in ancient ocean. It will stimulate our more deeply understanding about how Hg isotope would be used to trace paleo-redox conditions. I recommend accepting this manuscript in future after moderate revisions.

Introduction,

Line 37-44, I understand authors want to show Ediacaran biota evolution is very important. However, Hg isotope data during this period (574Ma-560Ma) was not well explained in this manuscript. I suggest that authors can focus on PZE at the beginning of introduction. For example, what is PZE, distributions, origins, impaction on biota, and what proxies have been used to evidence PZE, and so on. And then How can Hg isotope identify the occurrence of PZE, it is very important for this study.

Line 20, In the international stratigraphic chart (2015), the Ediacaran was dated as 635-542 Ma.

Line 53, Anoxic condition in fact include Ferruginous condition.

Line 58, Please specify "some data"

Line 61, Sponge could not be compared with Ediacaran biota, because sponge can live in a wide redox condition, however, Ediacaran biota would require more high oxygen level in seawater. This could not provide correct background of redox conditions.

Line 65, Could authors provide some evidence for a photic zone? Or some references would be required here. The marine photic euxinia is equal to the dynamic euxinic wedge reported in previous studies?

Line 187, I can understand organic materials are the main phase for Hg in sediments. But why does the absent of correlation between Hg and S indicate sulfide is a main host phase for Hg?

Line 229-230, Assuming all Hg came from seawater, it is right.

Line 263-264, Could authors show the whole range of $\Delta^{199}\text{Hg}/^{201}\text{Hg}$ of natural samples? Readers could not realize the difference between 1 and 1.5 is very large or small.

line 279-284, I think $\Delta^{199}\text{Hg}$ and $\delta^{202}\text{Hg}$ values should be the first-order evidence because 199/201 ratios are strongly dependent on sample numbers.

Line 315-316, Assuming this mechanism play important role in controlling the change of ^{199}Hg , but no correlations between S and Hg could also suggest that the photoreduction of Hg(II)-S is not the only process involved.

Line 323-325, In this experiment, how much S (H_2S or other species S) was conditioned.

Line 334, Again, sulfide could not be a main host phase for Hg base on the relationship between S and Hg. Did authors measure S concentrations in organic materials or organic S concentrations in samples? As such, the relationship between Hg and organic S may solid this interpretation.

Line 337-340, How did authors identify these processes, if Hg(0) oxidation in sulfidic waters is comparable to photooxidation and photoreduction in natural water environment?

Line 340-342, This recent study concludes that 50% to 76% of total atmospheric Hg input to the ocean mostly based on the change of $\Delta^{200}\text{Hg}$ isotope data of seawater from various modern oceans. However, $\Delta^{200}\text{Hg}$ signals of Doushantuo Formation is consistent.

Line 349-351, Fe species data indicate euxinic conditions between E intervals?

Line 351-361, I guess authors would like to state that euxinia recorded by Fe species data could be present in seawater column or porewater, but PZE recorded by negative ^{199}Hg only occurred in seawater column. It is why some euxinic sediments did not show negative ^{199}Hg . If it is true, authors could show more evidence from previous references.

line 384-386, How could author draw this conclusion? Please explain in detail.

Line 390-394, If the Nanhua basin is well connected with open ocean during OOE, why oxic seawater Tl, Mo isotope signals have not been preserved in the WH section.

Line 424-425, Authors want to state that P can be accumulated in anoxic but not euxinic conditions, however, previous Fe species data did not show a contracted euxinia.

Line 433-436, I hope authors can exactly sate OOE. Do OOE represent oxygenation of surface water in the Nanhua Basin or global seawater, or global deep water, or atmosphere? Tl isotopes indicate low oxygen level in open ocean? Near -2 Tl isotope could not support a more transiently and weakly oxygenated global ocean. Here authors should explain clearly.

Line 467, The weathering rate should be slow?

Line 469, Could authors tell readers where did nutrient element in open ocean come from?

Line 499-503, I'm confusing. Authors mentioned that oxygenation is favor to Ediacaran fossils, but both Weng'an and Wenghui biotas did not appear during OOE.

Line 511-512, What were new details and nuances evidenced by new Hg isotope data, please specify.

Line 518, I do not see that stable oceanic redox conditions were established by Hg isotope.

Line 520-522, The advantage of Hg isotope should be clearly stated in this manuscript. Previous studies identified three euxinic intervals in WH section, but Hg isotope only recognize two of them.

I hope these comments would be useful for improve this manuscript.

Reviewer #2 (Remarks to the Author):

The Ediacaran Period is a pivotal timeframe in the evolution of life on the earth, significant. This paper discussed mercury isotopes from multiple black shale sections of the Doushantuo Formation in South China to refine the reconstruction of the redox evolution for the Ediacaran ocean. A new redox proxy – mercury (Hg) isotope compositions is used to indicate recurrent photic zone euxinia limited ocean oxygenation and animal evolution in this paper. The finding is interesting. However, some questions need to be solved.

Three ocean oxygenation events were recorded by $\delta^{13}\text{C}$ and $\delta^{34}\text{S}_{\text{py}}$ in the WH section (E1, E2 and E3), and the enhanced continental weathering may transport large amounts of SO_4^{2-} to ocean, causing short-lived ocean oxygenation. $\Delta^{199}\text{Hg}$ shows near-zero in the three intervals in this paper may indicate volcanic materials, continental weathering materials and mixed materials. More evidence is needed to indicate $\Delta^{199}\text{Hg}$ is related to PZE rather than the above sources.

Compared with the $\Delta^{199}\text{Hg}$ reported in this paper, Zheng et al. (2018) reported a light value of $\Delta^{199}\text{Hg}$ from -0.22‰ ~ -0.12‰ . Thus, Hg isotope in this study associated with spatially dynamic PZE may have some controversy.

Reviewer #3 (Remarks to the Author):

Zheng et al. analyzed mercury concentrations and isotopes of four drill cores from South China Basin to investigate the geochemical records of the Ediacaran Doushantuo Formation. Samples analyzed include the entire WH section, and intervals of ocean oxygenation events (OOE) in the WA, TY and YJ sections. Based on mercury concentrations and isotopes, they discovered that mercury isotopic abnormalities parallel with OOE intervals and they thus suggest that OOE intervals represent time periods of photic zone euxinia (PZE). Given patterns of mercury isotopes along the shallow-deep basin transect, they discussed the putative spatial extent of PZE in the South China Basin. Authors further discussed plausible drivers and negative feedback of PZE, as well as the possible impacts of PZE on the Ediacaran biota. Overall, authors presented comprehensive and solid dataset on mercury concentrations and isotopes and the newly acquired data are consistent with patterns of other geochemical proxies reported previously. Based on my limited knowledge, I fee several points are relatively weakly supported or the interpretation is somehow against our current knowledge on PZE.

1) The hypothesis and argument lie heavily on a former study by the same author (Zheng et al., 2018), which proposed that mercury mass independent fractionation (MIF) is indicative of PZE. Nonetheless, El Mreiti and Atar groups are not typical environments of PZE, however, I do not deny that they may be highly euxinic. This statement is based on a related study by Gueneli et al (2018) that reported the discovery of aryl isoprenoids but not the detection of intact C40 carotenoids. To attribute mercury isotope MIF to the PZE, authors carried out extensive discussion on potential drivers of mercury MIF. Through this discussion, they tested the hypotheses of atmospheric precipitation, terrestrial flux, hydrothermal inputs and denied all of them due to one or several unexplainable facts. To the last, authors argued that PZE is the driver of patterns of mercury MIF in these four cores. However, this mechanism still involves the enhanced nutrient supply from the surrounding land and the deposition of atmospheric mercury and its further oxidation by reduced sulfur ligands. Therefore, enhanced terrestrial nutrient supply is essential to the development of PZE, mercury from terrestrial sources, as a possible driver of MIF, has been denied in this case. It is highly possible that MIF patterns could be fully explained when mercury inputs from terrestrial and atmosphere are both involved. Overall, I feel the attribution of mercury MIF pattern in relation to OOE intervals to the development of PZE is not precisely supported, or that more clear and robust arguments are needed to support this point.

2) The full Hg profile is only presented for the location of WH, whereas other three locations only presented Hg data for the OOE layers, but not the intervals between OOE. Without “background” data from other localities, it is hard to demonstrate whether Hg MIF patterns are synchronous across this basin, especially considering contrasting Hg isotopic signatures at OOE intervals between locations. Particularly, $\Delta^{199}\text{Hg}$ values at the three OOE intervals are near zero at the WH station with slight negative excursion, whereas values of $\Delta^{199}\text{Hg}$ are positive at WA and TY stations. Hg excursions have been used to argue for the development of PZE at OOE intervals, however, such excursions have only been supported for the WH station. I am not sure if it is feasible to include full Hg profiles for all four locations, but it will certainly increase the reliability of the arguments and conclusions.

3) Both FeHR/FeT and FePY/FeHR show decreasing patterns at OOE intervals, suggestive of less euxinic (but not completely PZE) conditions. This is consistent with the oxygenation of the water, which reduced the availability of sulfide and hence highly reactive iron and pyrite, but increased the fraction of iron oxides. At OOE intervals, pyrite is not the favored form of Fe precipitation, and thus hinders the development of euxinia. This is against the argument of PZE at OOE intervals.

4) The first three sections of the discussion are supported by data and echo the topic of this study, albeit the argument needs further validation. However, the rest three sections (I guess the main implications of this study; 5.4, 5.5 and 5.6) are a bit too far stretched, as the discussion of these three sections lie on the conclusion of PZE from former three sections. To my knowledge, these three sections barely provides new theories or discoveries to enrich our current knowledge on the Ediacaran environment, but mostly of speculation or synthesis of previous knowledge. I feel some of the information can be

provided in the introduction section, but their insertion in the discussion as the important aspects of implication is improper.

Figs. S5 and S6 are essential for the discussion and should be inserted in the main text.

REVIEWER COMMENTS (Response in blue text)

Reviewer #1 (Remarks to the Author):

Zheng and co-authors reported Hg concentration and isotope ratios in four black shale sections of the Ediacaran Doushantuo Formation in South China. And they confirmed widespread and spatially dynamic PZE on the continental margin of South China during at least two of the previously purported OOE intervals of the Ediacaran Period. The origins of PZE, both positive feedback on primary productivity and negative feedbacks that inhibited oxygen production, were discussed from regional and global scales. The study firstly provides more interesting and high-quality Hg isotope data during early-middle Ediacaran period, but also add some new knowledge in ancient ocean. It will stimulate our more deeply understanding about how Hg isotope would be used to trace paleo-redox conditions. I recommend accepting this manuscript in future after moderate revisions.

Response: We greatly appreciate that the reviewer acknowledges the novelty of this paper, and provides valuable suggestions. We have done our best to address these concerns. Please see our detailed responses to each comment below. We have also done numerous additional revisions to improve various aspects of the manuscript. Please see the revised manuscript for details. Below we numbered the comments as “C1-1, C1-2 ...” and our responses as “R1-1, R1-2 ...”.

Introduction,

C1-1: Line 37-44, I understand authors want to show Ediacaran biota evolution is very important. However, Hg isotope data during this period (574Ma-560Ma) was not well explained in this manuscript. I suggest that authors can focus on PZE at the beginning of introduction. For example, what is PZE, distributions, origins, impaction on biota, and what proxies have been used to evidence PZE, and so on. And then How can Hg isotope identify the occurrence of PZE, it is very important for this study.

R1-1: Thanks for the suggestion. We simplified the introduction on the evolution of Ediacara biota (the first paragraph of the introduction) and focus on the evolution of redox conditions and how Hg isotopes can provide new insights into the redox variation. We also largely simplified the discussion on the impact of PZE on Ediacara biota according to this comment (see our reply R1-25).

C1-2: Line 20, In the international stratigraphic chart (2015), the Ediacaran was dated as 635-542 Ma.

R1-2: According to the latest version of the stratigraphic chart (2022), the Ediacaran Period is 635-539 Ma. Please see the latest version in this link: <https://stratigraphy.org/chart#latest-version>

C1-3: Line 53, Anoxic condition in fact include Ferruginous condition.

R1-3: Deleted “Ferruginous”, as anoxic is a more general term describing both ferruginous and euxinic conditions (Line 47).

C1-4: Line 58, Please specify “some data”

R1-4: Rephrased this sentence to “It was even suggested that surface ocean environments that met the oxygen requirements of the earliest metazoans (0.5–4.0% of present atmospheric levels) were already established well before the Ediacaran” (Line 51-53).

C1-5: Line 61, Sponge could not be compared with Ediacaran biota, because sponge can live in a

wide redox condition, however, Ediacaran biota would require more high oxygen level in seawater. This could not provide correct background of redox conditions.

R1-5: Deleted “sponges”, but just use the more general term “animals”, which is consistent with the original word used in the cited reference (Mills et al., 2014) (Line 54).

C1-6: Line 65, Could authors provide some evidence for a photic zone? Or some references would be required here. The marine photic euxinia is equal to the dynamic euxinic wedge reported in previous studies?

R1-6: At line 65, we were just explaining the concept of photic zone euxinia (PZE), and two references are already provided at the end of this sentence. The PZE and the dynamic euxinic wedge reported in previous studies share some common features as they are both water column euxinic conditions, but they are not equal. The PZE emphasizes that the euxinic water mass reached the photic zone, whereas “euxinic wedge” could not distinguish if the euxinic water mass was within the photic zone or in much deeper water. The PZE is a more detrimental condition to early animals which are found to occupy shallow shelf areas¹, as the invasion of euxinic water mass into the photic zone would more severely limit the habitable space for early animals. That is why PZE rather than deep water euxinia is often proposed to be the kill mechanism during mass extinctions². Another important effect of PZE is that it causes negative feedback on ocean oxygenation by promoting anoxygenic photosynthesis and limiting the oxygenic photosynthesis (as explained in section 4.5), and this could be a previously unrecognized mechanism that was responsible for the transient nature of the Ediacaran ocean oxygenation events. This effect could not be caused by a deep water (below the photic zone) euxinic wedge, which does not promote anoxygenic photosynthesis.

Therefore, a key novelty and advancement of our study relative to the previous studies on “euxinic wedge” is that we provide clear evidence for recurrent PZE during the Ediacaran Period based Hg isotopes, which provides new insights into the mechanism for the termination of the transient ocean oxygenation events (due to the negative feedback of PZE on oxygenic photosynthesis) and the link between ocean redox variation and the evolution of early animals.

C1-7: Line 187, I can understand organic materials are the main phase for Hg in sediments. But why does the absent of correlation between Hg and S indicate sulfide is a main host phase for Hg?

R1-7: **We re-examined the correlation between Hg and typical host phases of Hg (i.e., organic matter, sulfide and clay minerals), and found there are actually quite strong correlations between Hg and sulfide during the OOE intervals in Member II and IV of the WH section.** The correlations were originally examined for the entire WH section, but we found it is more reasonable to examine the correlations for OOE and non-OOE intervals separately between the host phases could be different under different redox conditions. **The correlations between Hg and pyrite S are actually stronger than that with TOC in both OOE intervals (Supplementary Figure S1), which further supports our argument of PZE during OOE intervals. We re-wrote the result section (Line 164-208) and added the detailed analysis of correlation between Hg and host phases in the supplemental material (Supplementary Text S1):**

“The correlations between THg and typical host phases of Hg in sediments (i.e., organic matter, sulfide and clay minerals) are often examined to help determine the mechanism of Hg enrichment in sediments³. Since the host phases could be different under different redox conditions³, it is reasonable to examine the correlations between THg and host phases for OOE and non-OOE intervals separately (Supplementary Figure S1). For WH section, THg shows strong correlations with both total organic carbon (TOC) and pyrite S (S_{py}) in both Member II and IV OOE intervals, but the correlations with S_{py} ($R^2 = 0.77$ and 0.69 , respectively) are even stronger than those with

TOC ($R^2 = 0.47$ and 0.62 , respectively). However, the correlations with both TOC and S_{py} are very weak in the Member III OOE interval, partly due to the scarcity of data points in this interval. In the non-OOE intervals of WH section, THg still shows a significant correlation with TOC ($R^2 = 0.46$, $P < 0.001$), but only a weak correlation with S_{py} ($R^2 = 0.22$, $P < 0.001$). There are no correlations with Al (a typical proxy of clay mineral) in the entire WH section except in the Member III OOE interval, but the uncertainty of this correlation is high due to the limited number of data points and their large scattering. For TY and YJ sections, the correlations between THg and all host phases are mostly insignificant except a relatively good correlation between THg and TOC for TY ($R^2 = 0.41$, $P < 0.001$). In WA section, THg shows strong correlations with both TOC ($R^2 = 0.85$, $P < 0.001$) and Al ($R^2 = 0.83$, $P < 0.001$), but no correlation with S_{py} (Supplementary Figure S1).”

C1-8: Line 229-230, Assuming all Hg came from seawater, it is right.

R1-8: The possible sources of Hg to marine sediments include atmospheric deposition, terrestrial influx and submarine hydrothermal emission. All these sources would introduce Hg into the seawater, which then deposits to sediments via adsorbing to sinking organic or mineral particles. There is also no evidence of hydrothermal alternation to Doushantuo Formation, precluding the possibility of Hg loss or input due to hydrothermal fluids. Thus it is reasonable to propose that the Hg isotopes in sediments represent the primary Hg isotope signals of the contemporaneous seawater.

C1-9: Line 263-264, Could authors show the whole range of $\Delta^{199}\text{Hg}/\Delta^{201}\text{Hg}$ of natural samples? Readers could not realize the difference between 1 and 1.5 is very large or small.

R1-9: We added a comment here “typically between 1.0 and 1.1” (Line 269), and also added the reference line of atmospheric Hg(II) and Hg(0) in Supplementary Figure S4 to show that the $\Delta^{199}\text{Hg}/\Delta^{201}\text{Hg}$ of Doushantuo shales is significantly different from that of atmospheric Hg.

C1-10: line 279-284, I think $\Delta^{199}\text{Hg}$ and $\delta^{202}\text{Hg}$ values should be the first-order evidence because 199/201 ratios are strongly dependent on sample numbers.

R1-10: Agree. We moved the argument based on the $\Delta^{199}\text{Hg}$ and $\delta^{202}\text{Hg}$ values in front of the argument based on the $\Delta^{199}\text{Hg}/\Delta^{201}\text{Hg}$ slope.

C1-11: Line 315-316, Assuming this mechanism play important role in controlling the change of ^{199}Hg , but no correlations between S and Hg could also suggest that the photoreduction of Hg(II)-S is not the only process involved.

R1-11: We re-examined the correlation between Hg and typical host phases of Hg (i.e., organic matter, sulfide and clay minerals), and found there are actually quite strong correlations between Hg and sulfide during the OOE intervals in Member II and IV of the WH section. Please see our response **R1-7**.

C1-12: Line 323-325, In this experiment, how much S (H_2S or other species S) was conditioned.

R1-12: This experiment used low-molecular-weight thiol compounds as the surrogates of reduced sulfur ligands. The concentration of thiol compounds is $5 \mu\text{M}$ and the thiol/Hg rate is $\sim 100^4$. This relatively low ratio is chosen based on previous studies to reach a reasonable oxidation rate that allows to take samples every a few hours.

C1-13: Line 334, Again, sulfide could not be a main host phase for Hg base on the relationship

between S and Hg. Did authors measure S concentrations in organic materials or organic S concentrations in samples? As such, the relationship between Hg and organic S may solid this interpretation.

R1-13: We re-examined the correlation between Hg and typical host phases of Hg (i.e., organic matter, sulfide and clay minerals), and found there are actually quite strong correlations between Hg and sulfide during the OOE intervals in Member II and IV of the WH section. Please see our response **R1-7**.

C1-14: Line 337-340, How did authors identify these processes, if Hg(0) oxidation in sulfidic waters is comparable to photooxidation and photoreduction in natural water environment?

R1-14: In Zheng et al., 2013⁵, I measured the Hg(0) oxidation rate constant in the presence of various thiol compounds with a S/Hg ratio of 100, and found the rate constants are typically 0.1-0.3 h⁻¹ for most thiols. The rates of thiol-induced Hg(0) oxidation are comparable to those of other redox processes of Hg in the environment. For example, the rate constant of Hg(II) photoreduction typically ranges from ~0.28 to 0.65 h⁻¹ in natural freshwater. Abiotic dark reduction caused by reduced DOM has a rate between 0.57 and 5.52 h⁻¹, and microbial reduction mediated by *Geobacter sulfurreducens* PCA is reported to be ~0.30 h⁻¹. Photooxidation in surface waters shows rate constants between ~0.09 and 0.7 h⁻¹, and microbial oxidation mediated by *Desulfovibrio desulphuricans* ND132 has a rate of ~0.03 h⁻¹.

In actual sulfidic seawater, such as the Black Sea, the inorganic sulfide concentration can reach a few hundreds of μM and the organic sulfur can reach $\sim 1 \mu\text{M}$ via sulfurization of dissolved organic matter⁶. The typical total Hg concentration in Black Sea is $<10 \text{ pM}$ ⁷. Thus the S/Hg ratio is much higher than the ratio used in the experiment of Zheng et al., 2013. Thus the oxidation rate of Hg(0) in actual sulfidic seawater is expected to be even higher than the experimental values.

C1-15: Line 340-342, This recent study concludes that 50% to 76% of total atmospheric Hg input to the ocean mostly based on the change of $\Delta^{200}\text{Hg}$ isotope data of seawater from various modern oceans. However, $\Delta^{200}\text{Hg}$ signals of Doushantuo Formation is consistent.

R1-15: The reviewer is correct, and the almost invariant $\Delta^{200}\text{Hg}$ is actually supporting our argument that the cyclic variations of $\Delta^{199}\text{Hg}$ and $\delta^{202}\text{Hg}$ of WH section are not simply caused by changes in the proportions of atmospheric Hg(II) and Hg(0) deposition (which should cause cyclic variation of $\Delta^{200}\text{Hg}$ as well). I understand that the statement at line 340-342 may seem contradictory to the invariant $\Delta^{200}\text{Hg}$. However, the variation of Hg isotopes during PZE is not simply caused by the increased proportion of atmospheric Hg(0) deposition, but also caused by the isotope fractionation during the oxidation of Hg(0) in seawater. This oxidation process would result in an increase of $\delta^{202}\text{Hg}$ and decrease of $\Delta^{199}\text{Hg}$ (with an enrichment factor of -0.2‰ for $\Delta^{199}\text{Hg}$) in seawater due to nuclear volume effect, but would not produce $\Delta^{200}\text{Hg}$. Modern atmospheric gaseous Hg(0) typically shows negative $\Delta^{199}\text{Hg}$ ($\sim -0.20\text{‰}$) and slightly negative $\Delta^{200}\text{Hg}$ ($\sim -0.05\text{‰}$) (see Supplementary Text S3). If we assume the same isotope signatures for the atmospheric gaseous Hg(0) in the Ediacaran Period as the modern values, the fractionation process during Hg(0) oxidation in seawater would shift the $\Delta^{199}\text{Hg}$ to -0.40‰ without any change of $\Delta^{200}\text{Hg}$ for the isotope signatures of oxidized Hg(II) in seawater. Thus, it is possible to cause the observed variation of $\Delta^{199}\text{Hg}$ without significant changes of $\Delta^{200}\text{Hg}$.

To further elucidate my argument, we performed a simple calculation for the Hg isotope signatures of seawater caused by the mixing of atmospheric Hg(II) and Hg(0), taking into account the fractionation during Hg(0) oxidation in sulfidic water under PZE conditions, using a binary mixing model. The isotope signatures of atmospheric Hg(II) and Hg(0) are described in Supplementary Text S3. As explained above, the atmospheric Hg(0) undergoes a fractionation

under PZE conditions and thus its $\Delta^{199}\text{Hg}$ is shifted to -0.40‰ . The “ f ” is the fraction of each endmember. Under background condition (during the non-OOE intervals), the isotope signatures of the WH section (up to 0.19‰ for $\Delta^{199}\text{Hg}$ and 0.03‰ for $\Delta^{200}\text{Hg}$) can be simulated with a $f_{\text{Hg}(0)}$ up to ~ 0.4 , which is consistent the fraction of atmospheric $\text{Hg}(0)$ deposition in modern open ocean estimated by Jiskra et al., 2021⁸ based on the $\Delta^{200}\text{Hg}$ values of modern seawater. However, during the PZE intervals (here I use the Member II OOE interval as an example), to reach the observed negative shift of $\Delta^{199}\text{Hg}$ (down to -0.10‰) in WH section, the $f_{\text{Hg}(0)}$ needs to increase to 0.63 . This increase of $f_{\text{Hg}(0)}$ would only lower the seawater $\Delta^{200}\text{Hg}$ to 0.01‰ , which is only 0.04‰ lower than the $\Delta^{200}\text{Hg}$ of the non-OOE intervals, and is indistinguishable from the typical analytical uncertainty of MIF ($\pm 0.05\text{‰}$).

		$\Delta^{199}\text{Hg}$ (‰)	$\Delta^{200}\text{Hg}$ (‰)	f
Background (non-OOE intervals)	Atmospheric $\text{Hg}(0)$	-0.20	-0.05	0.40
	Atmospheric $\text{Hg}(\text{II})$	0.41	0.12	0.60
	Isotope signatures of seawater caused by mixing of atmospheric $\text{Hg}(\text{II})$ and $\text{Hg}(0)$	0.17	0.05	
PZE condition (OOE intervals)	Atmospheric $\text{Hg}(0)$	-0.40	-0.05	0.63
	Atmospheric $\text{Hg}(\text{II})$	0.41	0.12	0.37
	Isotope signatures of seawater caused by mixing of atmospheric $\text{Hg}(\text{II})$ and $\text{Hg}(0)$, but taking into account the fractionation during $\text{Hg}(0)$ oxidation in sulfidic surface seawater as a result of PZE	-0.10	0.01	

Since the original text at line 340-342 may cause confusion, and it is not essential for the discussion of the possible mechanism of Hg isotope fractionation during PZE, we deleted this statement in the revised manuscript, and clarified that the shift of $\Delta^{199}\text{Hg}$ in the second mechanism is driven by the isotope fractionation during thiol-induced $\text{Hg}(0)$ oxidation in seawater, rather than by the isotope signal of atmospheric $\text{Hg}(0)$ itself (Line 340-357).

C1-16: Line 349-351, Fe species data indicate euxinic conditions between E intervals?

R1-16: As the reviewer stated in the next comment, the Fe speciation data between E intervals likely indicates bottom water euxinia or sediment porewater euxinia, as suggested by Sahoo et al.⁹ who first reported these Fe speciation data.

C1-17: Line 351-361, I guess authors would like to state that euxinia recorded by Fe species data could be present in seawater column or porewater, but PZE recorded by negative 199Hg only occurred in seawater column. It is why some euxinic sediments did not show negative 199Hg. If it is true, authors could show more evidence from previous references.

R1-17: The reviewer is correct that Hg isotopes are sensitive to PZE, which is the presence of sulfidic water in the photic zone, but not necessarily sensitive to euxinia that developed only in deep water or in porewater. We rephased this part of the discussion according to the reviewer’s suggestion (Line 365-375):

“Sahoo et al. who first reported these Fe speciation data suggested that the high $\text{Fe}_{\text{py}}/\text{Fe}_{\text{HR}}$ could potentially reflect sulfidic conditions in sediment porewater rather than the water column¹⁰. In contrast, Hg isotopes specifically track euxinia in the photic zone, because the two proposed mechanisms controlling Hg isotope fractionation under PZE both operate in surface water where light can penetrate or atmospheric gaseous $\text{Hg}(0)$ can dissolve. There is currently no known mechanism that produces characteristic Hg isotope signatures in euxinic (but non-PZE) bottom

water or sediment porewater, where sulfide only acts as a ligand that binds with Hg. In fact, the study by Zheng et al. found that the deeper-water section deposited under euxinic (but not necessarily PZE) conditions shows no or a much weaker negative shift in $\Delta^{199}\text{Hg}$ than the shallower section where PZE occurred¹¹, suggesting that Hg isotopes track specifically PZE rather than euxinia in general.”

C1-18: line 384-386, How could author draw this conclusion? Please explain in detail.

R1-18: This is a speculation based on the fact that the E1 interval coincided with a sea level highstand, and the sea level was in general dropping after the initial highstand immediately after the Marinoan glaciation¹². Changes of sea level could change the depth of the chemocline and thus may affect the depth of the euxinic water mass. However, this argument is speculative and we added “although this hypothesis needs to be tested further” to clarify the uncertainty (Line 411).

C1-19: Line 390-394, If the Nanhua basin is well connected with open ocean during OOE, why oxic seawater Tl, Mo isotope signals have not been preserved in the WH section.

R1-19: The Tl and Mo isotope data of the WH section were reported in previous publications by Chad Ostrander^{12,13}, who is a co-author of the current paper. He proposed that Tl and Mo isotope signals could be explained by changes in the connectivity of the Nanhua Basin to the open ocean. However, the hypothesis of a partially restricted Nanhua Basin is still subject to debate and is inconsistent with the paleogeographic reconstruction. As we have pointed out in the manuscript, the connectivity of South China to the open ocean only affects the source of nutrients that triggered PZE while the mechanism that links PZE to enhanced nutrient influx remains the same. The focus and nuances of our study is to provide the first evidence of PZE in the Ediacaran ocean and its impact on ocean oxygenation and possibly the evolution of Ediacara biota. Thus whether the Nanhua Basin was restricted would not change our conclusion and is not a focus of this study. To avoid confusion to readers, we deleted the description on the debates over the connectivity of Nanhua Basin.

C1-20: Line 424-425, Authors want to state that P can be accumulated in anoxic but not euxinic conditions, however, previous Fe species data did not show a contracted euxinia.

R1-20: As explained in R1-17, the Fe speciation between the E intervals may actually indicate porewater euxinic condition rather than the water column euxinia according to Sahoo et al. (2012)¹⁰. Thus if the water column was non-euxinic between the E intervals, the Fe speciation data is not contradictory with the accumulation of P.

C1-21: Line 433-436, I hope authors can exactly state OOE. Do OOE represent oxygenation of surface water in the Nanhua Basin or global seawater, or global deep water, or atmosphere? Tl isotopes indicate low oxygen level in open ocean? Near -2 Tl isotope could not support a more transiently and weakly oxygenated global ocean. Here authors should explain clearly.

R1-21: OOE was referring to the oxygenation of global seawater when it was initially proposed by Sahoo et al. However, since our Hg isotope data does not directly indicate ocean oxygenation, we prefer to use “E” intervals instead of “OOE” in our manuscript, although we still refer to OOE in several cases in order to discuss the relationship between PZE and OOE. As we responded in R1-19, the interpretation of Tl isotopes relies on the hypothesis of a partially restricted Nanhua Basin, which is subject to debate and is not the focus of our current study. Therefore, we deleted the discussion on Tl isotopes to avoid confusion and distraction to readers.

C1-22: Line 467, The weathering rate should be slow?

R1-22: Here what we want to say is that the weathering rate is supposed to slow down once the freshly exposed, easily weathered land surface after glacial retreat was weathered, which is a reasonable speculation and has been invoked to explain the transient ocean oxygenation event after the Marinoan glaciation¹⁴.

C1-23: Line 469, Could authors tell readers where did nutrient element in open ocean come from?

R1-23: This statement is from the study by Laakso et al., (2020)¹⁵. This study found that P regenerated at depth in oceanic basins and subsequently upwelled outpaced the delivery of P from continents, sustaining the positive feedback on primary production. The P in open ocean still requires the input from continental sources, but this study highlights the important role of P remineralization, which provides more P than continental weathering alone and potentially sustained P supply after the surge of continental weathering. Our purpose to cite this study is to argue that the decrease in P influx due to decreased post-glacial weathering rate alone could not fully explain the transient nature of the ocean oxygenation event, because P from deep ocean could maintain the primary production. Thus, the negative feedback of PZE on oxygenic photosynthesis may have been a key mechanism for the transience of OOE.

C1-24: Line 499-503, I'm confusing. Authors mentioned that oxygenation is favor to Ediacaran fossils, but both Weng'an and Wenghui biotas did not appear during OOE.

R1-24: Please see **R1-25**. Since we did not directly measure samples from fossiliferous intervals. In the revised manuscript, we clearly stated this limitation (Line 499-500) and simplified the discussion on the biological impact.

C1-25: Line 511-512, What were new details and nuances evidenced by new Hg isotope data, please specify.

R1-25: We acknowledge that the current data could not provide a detailed evaluation of the impact of redox variations on the diversification of Ediacara biota, because we did not directly measure samples from fossiliferous intervals. In the revised manuscript, we clearly stated this limitation (Line 499-500) and simplified the discussion on the biological impact (which is also requested by reviewer 3). However, Hg isotope data of this study reveals the occurrence of PZE in the South China Basin and its negative feedback on oxygen production, which still provides new insights into the reason for the unstable redox conditions of the Ediacaran ocean. The unstable redox conditions and repeated development of PZE in early Ediacaran oceans as revealed by Hg isotopes in this study likely limited the ecological expansion of macroscopic multicellular eukaryotes, although more studies that compare fossiliferous and non-fossiliferous intervals are needed to further verify this hypothesis.

C1-26: Line 518, I do not see that stable oceanic redox conditions were established by Hg isotope.

R1-26: Please see **R1-25**. We removed this statement and shortened the discussion on the biological impact.

C1-27: Line 520-522, The advantage of Hg isotope should be clearly stated in this manuscript. Previous studies identified three euxinic intervals in WH section, but Hg isotope only recognize two of them.

R1-27: As we replied in **R1-17**, a key advantage of Hg isotopes over previous proxies is that Hg

isotopes specifically track PZE, which is the euxinia in the marine photic zone, not euxinia in general. Euxinia could develop in bottom water or in sediment porewater, which are not necessarily detrimental to shallow marine life, but PZE is particularly detrimental to shallow marine inhabitants and has been proposed as a potent kill mechanism during almost all Phanerozoic mass extinction events² and a key factor responsible for the evolutionary stasis during the mid-Proterozoic¹⁶. As we wrote in the introduction, “the shelf areas are believed to have been major habitats for early animals¹, and thus the development of PZE would have shrunk the oxygenated habitats and possibly contribute to the delayed rise and eventual decline of the Ediacara biota” (Line 63-66).

We now further clarified the novelty of the study at Section 4.5, Line 472-479: “Thus, we propose that the widespread PZE in continental margins initiated a negative feedback on oxygenic photosynthesis, which may be partially responsible for the relatively rapid termination of OOE. This argument is consistent with a recent hypothesis based on theoretical models that found the cyclic and oscillating oxygenation and deoxygenation events during the Ediacaran Period can be explained by internal feedbacks in the biogeochemical cycles of carbon, oxygen and phosphorous, without the need for specific external forcing¹⁷. **But as an important step forward, our finding of PZE provides an independent validation and new mechanistic insights for this theoretical hypothesis**” and in “Conclusion”, Line 504-508: “**Although the presence of dynamic “euxinic wedge” at South China during the Ediacaran Period has been reported, our study is the first time to show evidence for the invasion of euxinic water mass into the marine photic zone...**”.

I hope these comments would be useful for improve this manuscript.

Again, thank you so much for the detailed and valuable comments!

Reviewer #2 (Remarks to the Author):

The Ediacaran Period is a pivotal timeframe in the evolution of life on the earth, significant. This paper discussed mercury isotopes from multiple black shale sections of the Doushantuo Formation in South China to refine the reconstruction of the redox evolution for the Ediacaran ocean. A new redox proxy – mercury (Hg) isotope compositions is used to indicate recurrent photic zone euxinia limited ocean oxygenation and animal evolution in this paper. The finding is interesting. However, some questions need to be solved.

C2-1: Three ocean oxygenation events were recorded by $\delta^{13}\text{C}$ and $\delta^{34}\text{S}$ in the WH section (E1, E2 and E3), and the enhanced continental weathering may transport large amounts of SO_4^{2-} to ocean, causing short-lived ocean oxygenation. $\Delta^{199}\text{Hg}$ shows near-zero in the three intervals in this paper may indicate volcanic materials, continental weathering materials and mixed materials. More evidence is needed to indicate $\Delta^{199}\text{Hg}$ is related to PZE rather than the above sources.

R2-1: We greatly appreciate that the reviewer acknowledges the novelty of this paper, and provides valuable suggestions. We have done our best to address these concerns.

Regarding the comment on the possible influence of Hg sources on $\Delta^{199}\text{Hg}$, reviewer #3 asked the same question, and we have written a detailed reply in **R3-1** (the reply to reviewer 3’ first comment). **We have provided new evidence based on the correlations between Hg and host phases, as well as a simple box-model, which significantly strengthened our argument that the shifts of Hg isotopes during E intervals could not be fully explained by the mixing of Hg sources.** Below we summarize the main revisions that we have done to address this question:

1. **We re-examined the correlations between Hg concentration and typical host phases of Hg (i.e., organic matter, sulfide and clay minerals) and added a detailed analysis of these correlations in Supplementary Text S1 and Figure S1. We found there are actually quite strong correlations between Hg and sulfide during the OOE intervals in Member II and IV of the WH section ($R^2 = 0.77$ and 0.69 , respectively, $P < 0.001$), and they are notably stronger than the correlation with TOC ($R^2 = 0.47$ and 0.62 , respectively, $P < 0.001$). The strong correlation between Hg and sulfide is typically interpreted as a sign of euxinic depositional environment³, which provides further support for our argument of PZE during OOE intervals.** In contrast, between the OOE intervals, Hg shows a stronger correlation with TOC ($R^2 = 0.47$, $P < 0.001$) than with sulfide ($R^2 = 0.23$, $P = 0.002$), suggesting that the sedimentation of Hg was controlled by OM, like in modern open oceans¹⁸, consistent with our argument that PZE was contracted in the non-OOE intervals.

Importantly, the correlations of Hg with Al (often used as a proxy of continental weathering) are insignificant in the entire WH section, as well as in YJ and most intervals of TY sections. There was no terrestrial plants and soils during the Ediacaran Period. Thus the main form of Hg on land was geogenic Hg in crustal rocks. Thus the lack of correlation between Hg and Al suggests insignificant terrestrial Hg input. Only the WA section shows a significant correlation with both Al and TOC, but WA is the shallowest section located on the shelf margin and it is reasonable for it to receive more continental inputs. **Overall, the correlations of Hg with different host phases indicate that Hg was mainly associated with sulfide during OOE intervals, and with organic matter between OOE intervals, but was not associated with Al at any intervals. Thus terrestrial Hg was unlikely a major source of Hg. Other than Supplementary Text S1 and Figure S1, we also added these correlations to strengthen our arguments in multiple places in the manuscript (e.g. in Sections 4.1, 4.2 and 4.3).**

2. The Hg concentration in crustal igneous and metamorphic rocks is very low (typically $< 10\text{ppb}$)^{19,20}, much lower than the $> 100\text{ppb}$ (and frequently $> 1000\text{ppb}$) Hg in Doushantuo shales. Therefore, enhanced continental weathering is unlikely to supply enough Hg to significantly affect the isotope compositions of the ocean. This argument has been added to the revised manuscript (Section 4.1, Line 237-240).
3. **Evidence for possible volcanic input is only found in E3 (the Member IV OOE) based on simultaneous spikes of Hg concentration and Hg/TOC, Hg/S and Hg/Al, and we clearly discussed this possibility and acknowledged that the isotope signals in E3 may indicate volcanic source rather than PZE (although this does not necessarily mean PZE was absent in E3 because Hg still shows a strong correlation with sulfide and other redox proxies still suggests euxinic conditions in E3) (Line 441-456, and Supplementary Text S4).** Note that simultaneous increases of Hg concentration and the ratios of Hg vs. major host phases (rather than any of these values alone) are required to argue for volcanic inputs as suggested by Grasby et al., 2019. Thus it is clear that there was no evidence for volcanic inputs during E1 and E2 because either Hg concentration or the normalized ratios were not elevated during these two intervals.
4. To quantitatively estimate the effect of terrestrial weathering and local volcanism on Hg budget and $\Delta^{199}\text{Hg}$ in the ocean, **we constructed a simplified Hg isotope box-model for the studied time interval based on the previously published global Hg isotope box-model by Sun et al, 2019²¹. The details and results of this model are included in Supplementary Text S4, Figure S6 and Figure 4 (also attached here as Figure R1).** This model shows that even under a scenario of 50x increase of terrestrial or local volcanic Hg inputs, the $\Delta^{199}\text{Hg}$ of the ocean can only be decreased by $< 0.03\text{‰}$, which is an order of magnitude lower than the negative excursion of $\Delta^{199}\text{Hg}$ during E intervals (-0.19‰ to -0.29‰ , see

Section 3.2). The reason for the muted $\Delta^{199}\text{Hg}$ shift even under extremely high terrestrial and local volcanic Hg inputs might be that the enhanced terrestrial and volcanic inputs would increase the size of the oceanic Hg reservoir, which would also lead to enhanced re-emission of Hg from ocean to the atmosphere as observed in modern ocean^{21,22}. The re-emitted Hg would undergo atmospheric redox transformations that produce positive $\Delta^{199}\text{Hg}$ for atmospheric oxidized Hg(II) species²³, which then deposit back to land and ocean surfaces. Thus the enhanced re-emission and re-deposition of Hg with positive $\Delta^{199}\text{Hg}$ eventually counteracted the negative shift of $\Delta^{199}\text{Hg}$ caused by terrestrial and local volcanic Hg inputs. A full study on the effects of Hg re-emission and re-deposition is beyond the scope of the current study and is being investigated in a separate study. While this model is a simplified version of the more sophisticated model published by Sun et al, 2019, it is intended to serve as “additional” evidence to supplement the other evidence that are shown above. Therefore, **the modeling results further strengthen our argument that terrestrial or volcanic Hg could not fully account for the negative shift $\Delta^{199}\text{Hg}$ during E intervals. The above discussion has been added to the revised manuscript (Section 4.1, Line 240-256).**

Figure R1 (Figure 4 in the revised manuscript). Numerical simulation results using a simplified Hg isotope box-model. Simulated marine Hg enrichment (b) and $\Delta^{199}\text{Hg}$ (c) in response to an increase in terrestrial weathering Hg flux by 10–50× during E1 (~635 Ma), E2 (~580 Ma), and E3 (~565 Ma), and a further increase in local volcanic Hg emission by 100× during E3 (a). The “enrichment factor_ocean” value is defined as marine Hg reservoir relative to between-OOE reservoir size.

C2-2: Compared with the $\Delta^{199}\text{Hg}$ reported in this paper, Zheng et al. (2018) reported a light value of $\Delta^{199}\text{Hg}$ from -0.22‰ ~ -0.12‰. Thus, Hg isotope in this study associated with spatially dynamic PZE may have some controversy.

R2-2: We would like to clarify that:

1) **PZE would cause “negative shift” of $\Delta^{199}\text{Hg}$, but not necessarily “negative value” of $\Delta^{199}\text{Hg}$.** The “negative shift” refers to the change of $\Delta^{199}\text{Hg}$ relative to its background value. In our case, the background value (the $\Delta^{199}\text{Hg}$ during the non-OOE intervals) is quite positive (up to $\sim 0.2\text{‰}$), similar to those of modern open ocean sediments¹⁸. Thus the “negative shifts” of $\Delta^{199}\text{Hg}$ during the E intervals of WH section are already quite strong. E1 shows the strongest negative excursion of $\Delta^{199}\text{Hg}$ by as much as -0.29‰ (relative to the highest value between E intervals), whereas both E2 and E3 show smaller but still significant negative excursions by -0.19‰ . These negative shifts are indeed comparable to the $\sim -0.2\text{‰}$ shifts reported by Zheng et al. (2018). We have clarified this in the revised manuscript (Section 4.2, Line 185-187).

2) The negative excursions of $\Delta^{199}\text{Hg}$ during E intervals were accompanied by clear positive excursions of $\delta^{202}\text{Hg}$ with a characteristic $\Delta^{199}\text{Hg}/\delta^{202}\text{Hg}$ slope of ~ -0.1 . **These fractionation patterns are also fully consistent with Zheng et al. (2018)¹¹ as well as our new paper just accepted by EPSL (Zheng et al., 2023)²⁴.** These two publications reported Hg isotope fractionation for other time periods (Mesoproterozoic and Late Devonian) where PZE was evidenced independently by biomarkers. As we explained in the manuscript, these fractionation patterns (including the direction and extent of the shifts of both $\Delta^{199}\text{Hg}$ and $\delta^{202}\text{Hg}$ values as well as $\Delta^{199}\text{Hg}/\delta^{202}\text{Hg}$) are all consistent with the expected Hg isotope fractionation under PZE (Section 4.2). **It important to consider all these patterns of Hg isotopes together rather than $\Delta^{199}\text{Hg}$ alone, because together they provide “multi-dimensional” constraints on the mechanism of Hg isotope fractionation and thus allow us to pinpoint the processes driving these fractionations more accurately.** This is in fact how Hg isotopes are utilized to trace Hg cycling in modern studies²⁵. Thus we believe the link between Hg isotopes and PZE in this study is strongly supported by abundant and compelling evidence.

Reviewer #3 (Remarks to the Author):

Zheng et al. analyzed mercury concentrations and isotopes of four drill cores from South China Basin to investigate the geochemical records of the Ediacaran Doushantuo Formation. Samples analyzed include the entire WH section, and intervals of ocean oxygenation events (OOE) in the WA, TY and YJ sections. Based on mercury concentrations and isotopes, they discovered that mercury isotopic abnormalities parallel with OOE intervals and they thus suggest that OOE intervals represent time periods of photic zone euxinia (PZE). Given patterns of mercury isotopes along the shallow-deep basin transect, they discussed the putative spatial extent of PZE in the South China Basin. Authors further discussed plausible drivers and negative feedback of PZE, as well as the possible impacts of PZE on the Ediacaran biota. Overall, authors presented comprehensive and solid dataset on mercury concentrations and isotopes and the newly acquired data are consistent with patterns of other geochemical proxies reported previously. Based on my limited knowledge, I see several points are relatively weakly supported or the interpretation is somehow against our current knowledge on PZE.

Response: We greatly appreciate that the reviewer acknowledges the novelty of this paper, and provides valuable suggestions. We have done our best to address these concerns. Please see our detailed responses to each comment below. We have also done numerous additional revisions to improve various aspects of the manuscript. Please see the revised manuscript for details. Below we numbered the comments as “C3-1, C3-2 ...” and our responses as “R3-1, R3-2 ...”.

C3-1: The hypothesis and argument lie heavily on a former study by the same author (Zheng et al., 2018), which proposed that mercury mass independent fractionation (MIF) is indicative of PZE. Nonetheless, El Mreiti and Atar groups are not typical environments of PZE, however, I do not deny

that they may be highly euxinic. This statement is based on a related study by Gueneli et al (2018) that reported the discovery of aryl isoprenoids but not the detection of intact C40 carotenoids. To attribute mercury isotope MIF to the PZE, authors carried out extensive discussion on potential drivers of mercury MIF. Through this discussion, they tested the hypotheses of atmospheric precipitation, terrestrial flux, hydrothermal inputs and denied all of them due to one or several unexplainable facts. To the last, authors argued that PZE is the driver of patterns of mercury MIF in these four cores. However, this mechanism still involves the enhanced nutrient supply from the surrounding land and the deposition of atmospheric mercury and its further oxidation by reduced sulfur ligands. Therefore, enhanced terrestrial nutrient supply is essential to the development of PZE, mercury from terrestrial sources, as a possible driver of MIF, has been denied in this case. It is highly possible that MIF patterns could be fully explained when mercury inputs from terrestrial and atmosphere are both involved. Overall, I feel the attribution of mercury MIF pattern in relation to OOE intervals to the development of PZE is not precisely supported, or that more clear and robust arguments are needed to support this point.

R3-1: Thanks for raising this important question! **In the revised manuscript, we have provided new evidence based on the correlations between Hg and host phases, as well as a simple box-model, which significantly strengthened our argument that the shifts of Hg isotopes during E intervals could not be fully explained by the mixing of Hg sources.** Below we summarize the main revisions that we have done to address this question:

First, **we re-examined the correlations between Hg concentration and typical host phases of Hg (i.e., organic matter, sulfide and clay minerals) and added a detailed analysis of these correlations in Supplementary Text S1 and Figure S1.** These correlations are typically examined to help determine the mechanism of Hg enrichment in sediments³, which could help constrain the source of Hg and the depositional conditions. The correlations were originally examined for all samples of each section together, but we found it is more reasonable to examine the correlations for each OOE interval and non-OOE intervals separately between the host phases could be different under different redox conditions. **We found there are actually quite strong correlations between Hg and sulfide during the OOE intervals in Member II and IV of the WH section ($R^2 = 0.77$ and 0.69 , respectively, $P < 0.001$),** and they are notably stronger than the correlation with TOC ($R^2 = 0.47$ and 0.62 , respectively, $P < 0.001$), suggesting that while both organic matter and sulfide were important host phases of Hg during OOE intervals, sulfide played an even stronger role in the transport and sedimentation of Hg in ocean. **The strong correlation between Hg and sulfide is typically interpreted as a sign of euxinic depositional environment³, which provides further support for our argument of PZE during OOE intervals.** In contrast, between the non-OOE intervals, Hg shows a stronger correlation with TOC ($R^2 = 0.47$, $P < 0.001$) than with sulfide ($R^2 = 0.23$, $P = 0.002$), suggesting that the sedimentation of Hg was controlled by OM, like in modern open oceans¹⁸. Thus the change of host phases from sulfide to OM likely indicates a transition from locally euxinic to non- or less euxinic conditions, which is also consistent with our argument that PZE was contracted in the non-OOE intervals.

Also importantly, **the correlations of Hg with Al (often used as a proxy of continental weathering) are insignificant in the entire WH section, as well as in YJ and most intervals of TY sections.** Please note that there was no terrestrial plants and soils during the Ediacaran Period. Thus the main form of Hg on land was geogenic Hg in crustal rocks. Thus the lack of correlation between Hg and Al suggests insignificant terrestrial Hg input. Only the WA section shows a significant correlation with both Al and TOC, but WA is the shallowest section located on the shelf margin and it is reasonable for it to receive more continental inputs. **Overall, the correlations of Hg with different host phases indicate that Hg was mainly associated with sulfide during OOE intervals, and with organic matter between OOE intervals, but was not associated with Al at any intervals. Thus terrestrial Hg was unlikely a major source of Hg. Other than**

Supplementary Text S1 and Figure S1, we also added these correlations to strengthen our arguments in multiple places in the manuscript (in Sections 4.1, 4.2 and 4.3).

Second, besides the new evidence based on the correlations of Hg with host phases described above, we also added that the Hg concentration in crustal igneous and metamorphic rocks is very low (typically <10ppb)^{19,20}, which is much lower than the >100ppb (and frequently >1000ppb) Hg in Doushantuo shales. In contrast, phosphorous is a major element in upper continental crust (~0.1-0.2 wt%)²⁶. Therefore, while enhanced continental weathering can cause a surge of phosphorous influx to ocean, it is unlikely to supply enough Hg to significantly affect the isotope compositions of the ocean. **This argument has been added to the revised manuscript (Section 4.1, Line 237-240).**

Third, to quantitatively estimate the effect of terrestrial weathering and local volcanism on Hg budget and $\Delta^{199}\text{Hg}$ in the ocean, **we constructed a simplified Hg isotope box-model for the studied time interval based on the previously published global Hg isotope box-model by Sun et al, 2019²¹. The details and results of this model are included in Supplementary Text S4, Figure S6 and Figure 4 (also attached in this response as Figure R1).** This model shows that **even under a scenario of 50x increase of terrestrial or local volcanic Hg inputs, the $\Delta^{199}\text{Hg}$ of the ocean can only be decreased by <0.03‰, which is an order of magnitude lower than the negative excursion of $\Delta^{199}\text{Hg}$ during E intervals (-0.19‰ to -0.29‰, see Section 3.2).** The reason for the muted $\Delta^{199}\text{Hg}$ shift even under extremely high terrestrial and local volcanic Hg inputs might be that the enhanced terrestrial and volcanic inputs would increase the size of the oceanic Hg reservoir, which would also lead to enhanced re-emission of Hg from ocean to the atmosphere as observed in modern ocean^{21,22}. The re-emitted Hg would undergo atmospheric redox transformations that produce positive $\Delta^{199}\text{Hg}$ for atmospheric oxidized Hg(II) species²³, which then deposit back to land and ocean surfaces. Thus the enhanced re-emission and re-deposition of Hg with positive $\Delta^{199}\text{Hg}$ eventually counteracted the negative shift of $\Delta^{199}\text{Hg}$ caused by terrestrial and local volcanic Hg inputs. A full study on the effects of Hg re-emission and re-deposition is beyond the scope of the current study and is being investigated in a separate study. While this model is a simplified version of the more sophisticated model published by Sun et al, 2019, it is intended to serve as “additional” evidence to supplement the other evidence that are shown above. Therefore, **the modeling results further strengthen our argument that terrestrial or volcanic Hg could not fully account for the negative shift $\Delta^{199}\text{Hg}$ during E intervals. The above discussion has been added to the revised manuscript (Section 4.1, Line 240-256).**

At last, regarding the atmospheric deposition, it is the dominant Hg source to global ocean in the modern environment⁸, and thus was almost certainly a major Hg source in the Ediacaran ocean as well. We don't deny this fact at all. **Actually, we clearly pointed out that “while terrestrial geogenic Hg, atmospheric Hg, and deep ocean Hg-OM may all constitute the sources of Hg in the Doushantuo shales” (Line 294-297), but what we are arguing is that the variations of Hg isotope compositions could not be fully explained by changes of Hg sources, but require fractionation of Hg isotopes in ocean.** The atmospheric deposition could have supplied Hg to ocean, but this does not mean that sediments recorded the isotope signatures of atmospheric Hg sources without any fractionation. In fact, the two mechanisms we propose to fractionate Hg isotopes in PZE (photoreduction and oxidation of Hg(0)) are very common processes that Hg undergoes in modern aquatic environment, and both processes are experimentally quantified to cause characteristic Hg isotope fractionation with very similar patterns as observed in the OOE intervals (“pattern” means the direction and extent of both MDF and MIF values, as well as their relationships. Please see a detailed description of the fractionation patterns in Section 4.2 and 5.2). Thus, it is highly likely that the sediments were recording the isotope signatures modified by these fractionation processes rather than the original signatures of the sources. **This is the key**

hypothesis of this study, and it is supported by more than one previous studies. Other than Zheng et al., 2018, I have another paper just accepted by EPSL (Zheng et al., 2023)²⁴. This paper also found the same characteristic Hg isotope variations (negative shift of $\Delta^{199}\text{Hg}$ with concurrent positive shift of $\delta^{202}\text{Hg}$) due to PZE during the Late Devonian mass extinction, and the data of Zheng et al., 2023 is already presented in the current manuscript for a comparison (Figure 3).

We revised the manuscript based on the above argument to further clarify that, while the Hg in Doushantuo shales certainly came from its sources, the isotope compositions in sediments could not be simply interpreted as the mixing of different sources, because the original isotope signatures of atmospheric Hg(II) and Hg(0) sources can be significantly changed by Hg isotope fractionation in ocean (such as the photoreduction Hg(II)-S and the oxidation of atmospheric Hg(0) by thiol) after their deposition to ocean. PZE is the most likely mechanism that promotes these fractionation processes, and the pattern of Hg isotope fractionation during PZE has been well characterized by multiple studies (Section 4.2).

Figure R1 (Figure 4 in the revised manuscript). Numerical simulation results using a simplified Hg isotope box-model. Simulated marine Hg enrichment (b) and $\Delta^{199}\text{Hg}$ (c) in response to an increase in terrestrial weathering Hg flux by 10–50× during E1 (~635 Ma), E2 (~580 Ma), and E3 (~565 Ma), and a further increase in local volcanic Hg emission by 100× during E3 (a). The “enrichment factor_ocean” value is defined as marine Hg reservoir relative to between-OOE reservoir size.

C3-2: The full Hg profile is only presented for the location of WH, whereas other three locations only presented Hg data for the OOE layers, but not the intervals between OOE. Without “background” data from other localities, it is hard to demonstrate whether Hg MIF patterns are synchronous across this basin, especially considering contrasting Hg isotopic signatures at OOE intervals between locations. Particularly, $\Delta^{199}\text{Hg}$ values at the three OOE intervals are near zero at the WH station with

slight negative excursion, whereas values of $\Delta^{199}\text{Hg}$ are positive at WA and TY stations. Hg excursions have been used to argue for the development of PZE at OOE intervals, however, such excursions have only been supported for the WH station. I am not sure if it is feasible to include full Hg profiles for all four locations, but it will certainly increase the reliability of the arguments and conclusions.

R3-2: We totally agree that if the full profiles for the other sections can be measured, it will enhance our arguments, and we certainly wish to do so. However, the full-profile samples other than WH are not available, because the other sections were not completely preserved. For example, in WA and YJ sections, only the lower part is well preserved, whereas the upper parts (Member III and IV) were weathered. Over the last decade, several researchers have tried to replicate and find another well-preserved section, but so far, the Jiulongwan and the Wuhe sections are the most complete. For this reason, several recent publications for the same suite of samples also only reported the full profile for WH and the OOE intervals for other sections^{12,13,27}.

That being said, **the lack of the full profiles for other sections does not compromise the novelty and strength of our conclusion.** As indicated in this comment, the full profile of WH section shows clear, recurrent negative excursions of $\Delta^{199}\text{Hg}$, and we want to stress that these negative excursions of $\Delta^{199}\text{Hg}$ were accompanied by clear positive excursions of $\delta^{202}\text{Hg}$ with a characteristic $\Delta^{199}\text{Hg}/\delta^{202}\text{Hg}$ slope of ~ -0.1 . **It is important to consider all these patterns of Hg isotopes together rather than $\Delta^{199}\text{Hg}$ alone, because together they provide “multi-dimensional” constraints on the mechanism of Hg isotope fractionation and thus allow us to pinpoint the processes driving these fractionations more accurately.** This is in fact how Hg isotopes are utilized to trace Hg cycling in modern studies²⁵. As we already explained in the manuscript, these fractionation patterns (including the direction and extent of the shifts of both $\Delta^{199}\text{Hg}$ and $\delta^{202}\text{Hg}$ values as well as $\Delta^{199}\text{Hg}/\delta^{202}\text{Hg}$) are all consistent with the expected Hg isotope fractionation under PZE (Section 4.2). Moreover, these fractionation patterns are also surprisingly consistent with the Hg isotope fractionation reported in two previous studies for other time periods (Mesoproterozoic and Late Devonian) where PZE was evidenced independently by biomarkers (Figure 3). In addition, the strong correlation of THg with pyrite S during OOE intervals and the lack of correlation with pyrite S during non-OOE intervals (see details in our response **R3-1**) further support the argument for PZE during the previously proposed OOE intervals. Thus, the evidence for recurrent PZE in the WH section is abundant and very compelling.

Regarding the other sections, although full profiles are not available, all four sections have Hg isotope data for E1 (the Member II OOE interval) and they show clear variations of Hg isotopes within E1. By comparing the variation patterns of Hg isotopes during E1 across the four sections, which cover a transect from the shelf to the basin, we are able to evaluate the spatial extent of PZE and the dynamics of the euxinic water mass within the Ediacaran South China Basin. The E1 of WH also shows the strongest negative shift of $\Delta^{199}\text{Hg}$ among all E intervals, possibly indicating the most extensive PZE. Therefore, **our discussion for possible PZE in other sections was focusing on E1 interval only. We have further clarified this in the revised manuscript (Section 4.3, Line 377-385).**

Within E1, YJ (the basinal section) shows very similar Hg isotope patterns as the contemporaneous interval of WH, with negative $\Delta^{199}\text{Hg}$ down to -0.2‰ , positive $\delta^{202}\text{Hg}$ up to 0.5‰ , as well as a $\Delta^{199}\text{Hg}/\delta^{202}\text{Hg}$ of -0.09 . Thus it is reasonable to argue for PZE at YJ during this interval as well. Moreover, there is a clear positive shift of $\Delta^{199}\text{Hg}$ and negative shift of $\delta^{202}\text{Hg}$ going up the YJ section. Thus we argued that the PZE was the most extensive/persistent at the lower half and YJ and was weakening in the upper half (Section 4.3).

As for TY section (upper slope), although the absolute values of $\Delta^{199}\text{Hg}$ of TY are mostly slightly positive (up to 0.1‰), it is reasonable considering its shallower location. It likely received more

Hg input from the upwelling of Hg-OM (MIF >0), supported by the good correlation between THg and TOC for TY within E1 (see details in Section 4.1, Supplementary Text S1 and Figure S1). However, the patterns of Hg isotope variation within E1 at TY could not be explained by changes in Hg-OM source alone. There is a clear negative shift of $\Delta^{199}\text{Hg}$ and positive shift of $\delta^{202}\text{Hg}$ going up this interval, which are actually very similar to the variation patterns at WH within the same interval, as well as almost the same $\Delta^{199}\text{Hg}/\delta^{202}\text{Hg}$ (-0.09) as WH and YJ. These variations of Hg isotopes, if caused by changes in Hg-OM input, would indicate a decrease of Hg-OM input going up E1. However, both THg and TOC are increasing going up the E1 interval (see the **Figure R2** below), suggesting enhanced input of upwelling Hg-OM, contrary to the expectation based on Hg isotopes. Thus, changes in Hg sources could not fully explain the variation of Hg isotopes at TY within E1.

Instead, the mechanism for the variation patterns of Hg isotopes within E1 for the spatially diverse TY, WH and YJ sections are actually all connected, and they can all be well explained by the dynamic variation of the spatial coverage of PZE (the movement of euxinic water mass). In the revised manuscript, we strengthened the explanation for how the movement of euxinic wedge could explain the variations of Hg isotopes for TY, WH and YJ sections: “Furthermore, the euxinic wedge was becoming gradually shallower approaching the end of E1, and this is evident from the different patterns of Hg isotope variations within E1 across different sections. As discussed above, PZE was weakened at the deepest YJ section towards the end of E1, while PZE intensified at the shallower WH and TY sections, suggesting an up-slope migration of the euxinic water mass toward the end of E1 as depicted by Figure 6A” (Section 4.3, Line 403-408).

As for WA, it has the most positive $\Delta^{199}\text{Hg}$ within E1 among all sites. We clearly stated that “its THg correlates with TOC and Al, but not with sulfide (Supplementary Text S1 and Figure S1), suggesting that WA was likely located above the euxinic water mass or the PZE at WA was too sporadic to be captured by the current data. This is in agreement with previous paleogeographic reconstruction of the Doushantuo Formation in South China, which suggests that WA was likely exposed to oxic/suboxic water above the chemocline²⁸” (Section 4.3, Line 393-399).

Figure R2. The increases of THg and TOC going up the TY section within E1 interval.

C3-3: Both FeHR/FeT and FePY/FeHR show decreasing patterns at OOE intervals, suggestive of less

euxinic (but not completely PZE) conditions. This is consistent with the oxygenation of the water, which reduced the availability of sulfide and hence highly reactive iron and pyrite, but increased the fraction of iron oxides. At OOE intervals, pyrite is not the favored form of Fe precipitation, and thus hinders the development of euxinia. This is against the argument of PZE at OOE intervals.

R3-3: The Fe speciation during OOE intervals is actually quite variable, showing a wide range of values that indicate fluctuation between oxic and euxinic conditions. In contrast, Fe speciation seems to indicate persistently euxinic conditions during non-OOE intervals. Thus Fe speciation seems to be contradictory to our argument of PZE during OOE based on Hg isotopes. However, **we certainly noticed this issue and explicitly discussed the possible reasons (Section 4.2, Line 363-375).**

As we explained in details in responses **R3-1 and R3-2**, the evidence for PZE during OOE intervals (including all patterns of Hg isotope variations, the correlation between THg and pyrite S, and the similar variations of Hg isotopes as those previously reported for other sediments formed under PZE) are abundant and compelling. During non-OOE intervals, the opposite shifts of Hg isotopes and the lack of correlation between THg and pyrite S indicate weakened PZE. **The reason that Fe speciation seems to show the opposite might be that it is typically interpreted to reflect redox conditions at local bottom water rather than in the photic zone²⁹.** Sahoo et al. who first reported these Fe speciation data suggested that the high Fe_{py}/Fe_{HR} during non-OOE intervals could potentially reflect sulfidic conditions in sediment porewater rather than the water column¹⁰. **In contrast, Hg isotopes specifically track euxinia in the photic zone, because the two proposed mechanisms controlling Hg isotope fractionation under PZE both operate in surface water where light can penetrate or atmospheric gaseous Hg(0) can dissolve.** There is currently no known mechanism that produces characteristic Hg isotope signatures in euxinic (but non-PZE) bottom water or sediment porewater, where sulfide only acts as a ligand that binds with Hg. In fact, in Zheng et al. who first linked Hg isotopes to PZE, the deeper-water Atar Group deposited under euxinic (but not necessarily PZE) conditions according to its Fe speciation data, and it shows no or a much weaker negative shift of $\Delta^{199}Hg$ than the shallower El Mreiti Group where PZE occurred¹¹, suggesting that Hg isotopes track specifically PZE rather than euxinia in general.

C3-4: The first three sections of the discussion are supported by data and echo the topic of this study, albeit the argument needs further validation. However, the rest three sections (I guess the main implications of this study; 5.4, 5.5 and 5.6) are a bit too far stretched, as the discussion of these three sections lie on the conclusion of PZE from former three sections. To my knowledge, these three sections barely provides new theories or discoveries to enrich our current knowledge on the Ediacaran environment, but mostly of speculation or synthesis of previous knowledge. I feel some of the information can be provided in the introduction section, but their insertion in the discussion as the important aspects of implication is improper.

R3-4: Thanks for this suggestion. **We have largely simplified the discussion of the last three sections and combined the last two sections into a single Section 4.5. We also greatly strengthened the discussion in section 4.1, 4.2 and 4.3 and provided further validation for PZE (see details in response R3-1 and R3-2).** In this way, our paper is much more focused.

However, we don't completely agree that the last three sections (now Section 4.4 and 4.5) "barely provides new theories" or are "mostly synthesis of previous knowledge". Section 4.4 discussed the possible link between PZE and the enhanced nutrient input from post-glacial continental weathering, which is not only necessary for understanding the trigger of PZE, but also provides new insights into the interactions between various environmental change events and their consequences, since this study is the first report of PZE during the Ediacaran.

Section 4.5 highlights the novelty and impacts of this study. The negative impact of PZE on ocean oxygenation, although proposed previously for other time periods of Earth's history^{30,31}, has never been proposed for the Ediacaran Period. In fact, the mechanism for the transient oxygenation of ocean during the Ediacaran Period is important for understanding the course of Earth surface oxygenation, but is not yet clear. A recent hypothesis based on theoretical models that found the cyclic and oscillating oxygenation and deoxygenation events during the Ediacaran Period can be explained by internal feedbacks in the biogeochemical cycles of carbon, oxygen and phosphorous, without the need for specific external forcing¹⁷. **Our finding of PZE provides an independent validation and new mechanistic insights for this theoretical hypothesis, which is an important step forward (Line 477-479).** As we pointed in the manuscript that **“This may be an underappreciated mechanism for the transient nature of the Ediacaran OOs” (Line 460).**

The potential impact of PZE on the Ediacara biota is also important. However, we acknowledge that the current data could not provide a detailed evaluation of the impact of redox variations on the diversification of Ediacara biota, because we did not directly measure samples from fossiliferous intervals. **In the revised manuscript, we clearly stated this limitation (Line 497-500) and simplified the discussion on the biological impact.** The unstable redox conditions and repeated development of PZE in early Ediacaran oceans as revealed by Hg isotopes in this study likely limited the ecological expansion of macroscopic multicellular eukaryotes, although more studies that compare fossiliferous and non-fossiliferous intervals are needed to further verify this hypothesis.

C3-5: Figs. S5 and S6 are essential for the discussion and should be inserted in the main text.

R3-5: Thanks for this suggestion. We moved Figure S6 into the main text (now Figure 5). However, we still would like to leave Figure S5 in the supplementary material because it only contains Hg isotope data from the literature and the average values of main Hg sources have already been plotted on Figure 3. We also added a new Figure 4 (the simulation results of the Hg isotope box-model). So there are already 6 figures in the main manuscript.

Reference:

1. Tostevin, R. *et al.* Low-oxygen waters limited habitable space for early animals. *Nat Commun* **7**, 1–9 (2016).
2. Whiteside, J. H. & Grice, K. Biomarker records associated with mass extinction events. *Annu. Rev. Earth Planet. Sci.* **44**, 581–612 (2016).
3. Shen, J. *et al.* Sedimentary host phases of mercury (Hg) and implications for use of Hg as a volcanic proxy. *Earth Planet. Sci. Lett.* **543**, 116333 (2020).
4. Zheng, W. *et al.* Mercury stable isotope fractionation during abiotic dark oxidation in the presence of thiols and natural organic matter. *Environ. Sci. Technol.* **53**, 1853–1862 (2019).
5. Zheng, W., Lin, H., Mann, B. F., Liang, L. & Gu, B. Oxidation of dissolved elemental mercury by thiol compounds under anoxic conditions. *Environ. Sci. Technol.* **47**, 12827–12834 (2013).
6. Gomez-Saez, G. V. *et al.* Sulfurization of dissolved organic matter in the anoxic water column of the Black Sea. *Science Advances* **7**, eabf6199 (2021).
7. Lamborg, C. H. *et al.* Vertical distribution of mercury species at two sites in the Western Black Sea. *Marine Chemistry* **111**, 77–89 (2008).
8. Jiskra, M. *et al.* Mercury stable isotopes constrain atmospheric sources to the ocean. *Nature* **597**, 678–682 (2021).
9. Sahoo, S. K. *et al.* Oceanic oxygenation events in the anoxic Ediacaran ocean. *Geobiology* **14**, 457–468 (2016).
10. Sahoo, S. K. *et al.* Ocean oxygenation in the wake of the Marinoan glaciation. *Nature* **489**, 546–549 (2012).
11. Zheng, W., Gilleaudeau, G. J., Kah, L. C. & Anbar, A. D. Mercury isotope signatures record photic zone euxinia in the Mesoproterozoic ocean. *Proc. Natl. Acad. Sci. U.S.A.* **115**, 10594–10599 (2018).
12. Ostrander, C. M. *et al.* Multiple negative molybdenum isotope excursions in the Doushantuo Formation (South China) fingerprint complex redox-related processes in the Ediacaran Nanhua Basin. *Geochimica et Cosmochimica Acta* **261**, 191–209 (2019).
13. Ostrander, C. M. *et al.* Thallium isotope ratios in shales from South China and northwestern Canada suggest widespread O₂ accumulation in marine bottom waters was an uncommon occurrence during the Ediacaran Period. *Chemical Geology* **557**, 119856 (2020).
14. Lenton, T. M., Boyle, R. A., Poulton, S. W., Shields-Zhou, G. A. & Butterfield, N. J. Co-evolution of eukaryotes and ocean oxygenation in the Neoproterozoic era. *Nature Geosci* **7**, 257–265 (2014).
15. Laakso, T. A., Sperling, E. A., Johnston, D. T. & Knoll, A. H. Ediacaran reorganization of the marine phosphorus cycle. *PNAS* **117**, 11961–11967 (2020).
16. Johnston, D. T., Wolfe-Simon, F., Pearson, A. & Knoll, A. H. Anoxygenic photosynthesis modulated Proterozoic oxygen and sustained Earth’s middle age. *Proceedings of the National Academy of Sciences* **106**, 16925–16929 (2009).
17. Alcott, L. J., Mills, B. J. W. & Poulton, S. W. Stepwise Earth oxygenation is an inherent property of global biogeochemical cycling. *Science* **366**, 1333–1337 (2019).
18. Grasby, S. E., Them, T. R., Chen, Z., Yin, R. & Ardakani, O. H. Mercury as a proxy for volcanic emissions in the geologic record. *Earth Sci Rev* **196**, 102880 (2019).
19. Deng, C. *et al.* Mercury isotopic compositions of the Precambrian rocks and implications for tracing mercury cycling in Earth’s interior. *Precambrian Research* **373**, 106646 (2022).
20. Deng, C. *et al.* Mercury isotopic composition of igneous rocks from an accretionary orogen: Implications for lithospheric recycling. *Geology* (2022) doi:10.1130/G50131.1.
21. Sun, R. *et al.* Modelling the mercury stable isotope distribution of Earth surface reservoirs: Implications for global Hg cycling. *Geochim. Cosmochim. Acta* **246**, 156–173 (2019).

22. Zhang, Y. *et al.* An updated global mercury budget from a coupled atmosphere-land-ocean model: 40% more re-emissions buffer the effect of primary emission reductions. *One Earth* **6**, 316–325 (2023).
23. Song, Z., Sun, R. & Zhang, Y. Modeling mercury isotopic fractionation in the atmosphere. *Environ. Pollut.* **307**, 119588 (2022).
24. Zheng, W. *et al.* Mercury isotope evidence for recurrent photic-zone euxinia triggered by enhanced terrestrial nutrient inputs during the Late Devonian mass extinction. *Earth and Planetary Science Letters* (Accepted).
25. Blum, J. D., Sherman, L. S. & Johnson, M. W. Mercury isotopes in Earth and environmental sciences. *Annu. Rev. Earth Planet. Sci.* **42**, 249–269 (2014).
26. Hu, Z. & Gao, S. Upper crustal abundances of trace elements: A revision and update. *Chemical Geology* **253**, 205–221 (2008).
27. Xu, D. *et al.* Chromium isotope evidence for oxygenation events in the Ediacaran ocean. *Geochimica et Cosmochimica Acta* **323**, 258–275 (2022).
28. Jiang, G., Shi, X., Zhang, S., Wang, Y. & Xiao, S. Stratigraphy and paleogeography of the Ediacaran Doushantuo Formation (ca. 635–551Ma) in South China. *Gondwana Research* **19**, 831–849 (2011).
29. Raiswell, R. *et al.* The iron paleoredox proxies: A guide to the pitfalls, problems and proper practice. *Am J Sci* **318**, 491–526 (2018).
30. Planavsky, N. J. *et al.* Evolution of the structure and impact of Earth’s biosphere. *Nat Rev Earth Environ* **2**, 123–139 (2021).
31. Sánchez-Baracaldo, P., Bianchini, G., Wilson, J. D. & Knoll, A. H. Cyanobacteria and biogeochemical cycles through Earth history. *Trends in Microbiology* **30**, 143–157 (2022).

REVIEWERS' COMMENTS

Reviewer #1 (Remarks to the Author):

I carefully read this revised paper; most issues have been addressed by authors. I have some minor concerns would be considered.

Line 60 or 81-83, Like the response (R1-6). Authors should point to the difference between PZE and traditional euxinic wedge, because iron species have indicated euxinic conditions at different sections studied here. So that the advantage of this paper would be highlighted.

Line 236-237, Please check if there are hiatus before the occurrence of E intervals.

Line 239, please focus on total Hg during E intervals

Line 260-262, Hg isotopes of Ediacaran Hg(II) and Hg(0) could be different from that of modern atmosphere, because Ediacaran atmospheric chemistry maybe not similar to modern.

Line 340-346, Here, authors highlight the oxidation of Hg(0) by the abundant thiol compounds under PZE conditions. If it is possible that surface water overlain the PZE had also become more oxic condition, where Hg(0) could also be oxidized. This is just a suggestion. E Figure 6, In this Figure, the PZE is seems to have been expanded. In fact, TY, WH and YJ are not so far, therefore PZE recognized in the Nahua Basin could be located in a small area, rather than a very supper large region.

Reviewer #2 (Remarks to the Author):

The authors have made the changes suggested.

Reviewer #3 (Remarks to the Author):

Zheng et al have carefully addressed my comments and revised the manuscript accordingly. I do not have any further comments. The current manuscript is properly written and represents an important contribution to the field. --Xingqian

REVIEWER COMMENTS (Response in blue text)

Reviewer #1 (Remarks to the Author):

I carefully read this revised paper; most issues have been addressed by authors. I have some minor concerns would be considered.

Response: Thank you!

Line 60 or 81-83, Like the response (R1-6). Authors should point to the difference between PZE and traditional euxinic wedge, because iron species have indicated euxinic conditions at different sections studied here. So that the advantage of this paper would be highlighted.

Response: Agree. We added “Different from the previously proposed “euxinic wedge” that refers to a mid-depth euxinic watermass, PZE emphasizes the shallowing of the euxinic watermass into the photic zone, which is particularly detrimental to shallow marine inhabitants...” (L59-61).

Line 236-237, Please check if there are hiatus before the occurrence of E intervals.

Response: The WH section is continuous and well preserved. As we already pointed out in the “Method” section (L452-454), there is a regional stratigraphic discontinuity at the Member II-III boundary (before E2), which tracks a widespread marine regression. This discontinuity could be time-equivalent with the Gaskiers glaciation (~580 Ma).

Line 239, please focus on total Hg during E intervals

Response: We added “in almost all Doushantuo shales” at L165. This includes both E intervals and the intervals between them, and it indicates that terrestrial Hg was a minor source throughout the entire Doushantuo sections.

Line 260-262, Hg isotopes of Ediacaran Hg(II) and Hg(0) could be different from that of modern atmosphere, because Ediacaran atmospheric chemistry maybe not similar to modern.

Response: While it is hard to actually tell if the isotope signatures of atmospheric Hg(II) and Hg(0) in the Ediacaran Period were the same as the modern ones or not, the published data on Hg isotope signatures of Ediacaran and Cambrian marine sediments that deposited under oxic water and background Hg loading (without intensive volcanic inputs) are similar to modern open ocean sediments^{1,2}. Since the Hg isotope signatures of modern ocean sediments are driven by the mixed inputs of atmospheric Hg(II) and Hg(0), it is thus reasonable to hypothesize similar isotope signatures for the atmospheric Hg during the Ediacaran Period.

Line 340-346, Here, authors highlight the oxidation of Hg(0) by the abundant thiol compounds under PZE conditions. If it is possible that surface water overlain the PZE had also become more oxic condition, where Hg(0) could also be oxidized. This is just a suggestion.

Response: It is unlikely that the surface water overlain the PZE has become “more oxic”. When PZE occurs, H₂S in the euxinic watermass would also migrate upward and invade into the overlain surface water as well as the atmosphere³. Thus during PZE, the overlain water should become less oxic.

E Figure 6, In this Figure, the PZE is seems to have been expanded. In fact, TY, WH and YJ are not so far, therefore PZE recognized in the Nahua Basin could be located in a small area, rather than a very supper large region.

Response: The PZE in Figure 6 is only intended for illustration purpose, not to scale with the actual size of the euxinic watermass. It is only intended to show that the PZE covered TY, WH and YJ sections in a dynamic way. The basin in Figure 6 is only a small area that just extends from the continental shelf, but not far into the open ocean. Thus the PZE depicted in figure 6 is in fact only covering the area near the continental shelf. It seems large just because the region where the four study sections are located is zoomed in, which is necessary to clearly show all events happening along with the expansion and contraction of PZE in the region.

Reviewer #2 (Remarks to the Author):

The authors have made the changes suggested.

Response: Thank you!

Reviewer #3 (Remarks to the Author):

Zheng et al have carefully addressed my comments and revised the manuscript accordingly. I do not have any further comments. The current manuscript is properly written and represents an important contribution to the field. –Xingqian

Response: Thank you!

Reference:

1. Wu, Y. *et al.* Global Hg cycle over Ediacaran–Cambrian transition and its implications for environmental and biological evolution. *Earth and Planetary Science Letters* **587**, 117551 (2022).
2. Fan, H. *et al.* Mercury isotopes track the cause of carbon perturbations in the Ediacaran ocean. *Geology* **49**, 248–252 (2021).
3. Meyer, K. M., Kump, L. R. & Ridgwell, A. Biogeochemical controls on photic-zone euxinia during the end-Permian mass extinction. *Geol* **36**, 747 (2008).